# Battery Electric Bus Network: Efficient Design and Cost Comparison of Different Powertrains

**Orlando Barraza \* and Miquel Estrada \***

Barcelona Innovative Transportation Research Group, Barcelona School of Civil Engineering,
Universitat Politècnica de Catalunya, 08034 Barcelona, Spain
**\*** Correspondence: orlando.marath.barraza@upc.edu (O.B.); miquel.estrada@upc.edu (M.E.)

**Abstract:** Recent advances in the electromobility of bus fleets improve transit service sustainability but require the development of vehicle recharging facilities. The paper defines a methodology to design an efficient transit network operated by battery electric buses in cities with grid-shaped road network, based on continuous approximations. An analytical model defines the optimal network configuration that minimizes the agency cost, the monetization of emissions and the travel time of transit users. The analytical model allows the comparison of total cost, emissions and bus performance to other fuel powertrains. The methodology is tested in Guadalajara (Mexico) to propose an alternative bus configuration, outperforming the current bus service and reducing the agency cost and environmental impact. The analytical model justifies the network rationalization in fewer routes to reduce the total cost of the system. The deployment of standard battery electric buses with opportunity charging scheme obtains the lowest total cost of the system.

**Keywords:** bus network design; battery electric bus; opportunity charging; emissions; life cycle analysis

## 1. Introduction

This paper aims to design an optimal bus network configuration operated by full electric powertrain, estimating the total number of electric buses, charging resources and the total cost incurred by transit agency and users. The proposed methodology allows a comparative analysis of the resulting bus design with internal combustion engine bus powertrains. The paper also checks the potential benefits and cost savings of the optimized bus network design obtained by diesel powertrains with the current bus network configuration in Guadalajara, Mexico. A sensitivity analysis is also performed with regard to the unit cost parameters in order to estimate under which domains of distance cost, resource depreciation cost, emission monetization and charging infrastructure, the battery-electric buses are more cost-efficient than conventional diesel buses.

Mass transit systems must provide a competitive performance to achieve a positive modal redistribution from private cars. To that end, several cities have undergone a significant improvement in the transit service, together with access restrictions to private vehicles, land use planning and redensification strategies. A crucial issue is to provide an efficient network with an outstanding performance at a limited operating cost. [1]. The optimal balance is achieved by the efficient provision of the spatial configuration of routes and the temporal coverage of services (headways). Therefore, the transit network design problem is aimed at minimizing the total cost incurred by users and transit agencies. This problem has received the attention of many researchers, proposing discrete or continuous-based optimization models.

Discrete-based optimization models need the construction of a graph to resemble the existing road configuration with which the bus network may operate. Given an origin–destination matrix (O–D), different mathematical optimization techniques have been proposed to solve the network design problem. This modeling approach allows

representing with a good detail the street configuration of the city and the demand distribution. Nevertheless, as this problem is known to be Np-Hard in computational time [2], mathematical models can only obtain exact solutions in small problems [3]. For medium and large networks, researchers proposed heuristic algorithms [4–9] to obtain a suboptimal solution in affordable computational time. Later, metaheuristics algorithms [10–14] have been developed to fine tune a given suboptimal solution.

On the other hand, continuous or parsimonious models propose ideal conceptualizations of transit network whose physical layout and temporal performance are controlled by a small set of spatial and temporal decision variables. They estimate the network performance in terms of user travel time and operating costs. Each system cost component incurred by users (in-vehicle, waiting and access times) or transit agency is formulated as a function of decision variables.

A set of compact formulas is developed for each cost term using geometric probability. Then, the combination of optimal decision variable values that minimize the total cost is obtained by a continuous optimization, following a grid search procedure. This set of optimal values allows the creation of the ideal skeleton of routes and stops as well as the frequency scheme at which the system needs to operate. Later, decision makers create the final map of the routes based on the available street network, resembling the ideal scheme as much as possible. The main advantage of these models is the low computational effort to obtain the solution and the wide transferability to numerous case instances where the city shape is similar to the street network pattern considered in the model.

Nevertheless, assumptions such as the uniform demand distributed must be considered to simplify the calculations in the models. Several contributions in this field were developed for grid [15], radial network configurations [16]. Daganzo [17] proposed hybrid configurations to serve square-shaped cities. In this approach, the potentialities of both grid and radial schemes were combined in the same network pattern. This scheme was extended to a hybrid concept to cities where the road network follows a circular and ring mesh of streets [18,19]. Other works have been done comparing the performance of door-to-door and transfer-based grid and hybrid networks, stating that the latter provides lower cost designs to serve the mobility of the city when demand is consolidated in the city center [20].

However, few authors have integrated the environmental impacts derived from the bus service in the network optimization. Chester et al. and Griswold [21,22] consider the greenhouse gas (GHG) contribution of the transit vehicle manufacturing, road, station construction and the fuel consumption in the network design problem. These emissions are integrated into the objective function and the new optimal values of network decision variables are evaluated for different transport modes (bus, BRT, LRT and metro). This approach has also been analyzed in Cheng et al. and Griswold et al. [23,24] in San Francisco and in the new Barcelona bus network proposed in Estrada et al. [25] respectively. In Griswold et al. [26], travel time elasticities are considered in transit and private car systems to evaluate the modal split changes in scenarios, where the total amount of GHG emissions for the transit operator or the total system is constrained.

In Cheng et al. [27], the optimal design and emissions of a more sophisticated transit system, composed by trunk and feeder services, is evaluated under an elastic demand assumption. It implies a competition between the private car and Internal Combustion Engine (ICE) transit services. Lee and Madanat [28] proposed a methodology to define the optimal network of the charging infrastructure to minimize the emissions in a given city, promoting full electric vehicle operation. Nevertheless, the proposed charging network configuration is developed for passenger cars moving in different areas of the city, without any link with existing bus corridors. The location of the charging facility is not incorporated to the transit network design problem.

One of the crucial issues in the former references addressing the transit network design was the estimation of the amount of GHG emissions generated in the operation, construction and installation of transit facilities and auxiliary resources. The emissions

generated during the vehicle operation due to fuel combustion (exhaust emissions), tire rolling, braking or fuel evaporation have been fully analyzed in the EEA model and Argone GREET model [29,30]. Moreover, several studies have conducted life cycle cost analysis (LCCA) of the batteries and vehicles including energy and environmental aspects, and real energy consumption of electric buses and its benefits in urban areas [31–38].

Nevertheless, the LCCA of elements not related to the bus operations (lane infrastructure and charging infrastructure) has been barely addressed in the literature. These parameters have also been complemented by the energy consumption and emission data gathered from demonstrations conducted in the ZeEUS project, one of the main studies devoted to test electric bus performance on real conditions [39]. Unfortunately, the vast majority of LCCA analysis highlighting the lower emission contribution of electric rolling resources, do not consider the extra number of resources that electric bus operations will cause. Therefore, the main motivation of this paper is to fill this gap. Hence, in this work an estimation for the number of electric resources and capital and operating cost that transit agencies will incur.

Section 2 describes the methodology and row data used to estimate the agency, user and environmental costs developed as a function of bus service design variables. These formulas are merged into an optimization model to identify the best bus network operation under different scenarios. Section 3 shows the result of the model presenting the optimal parameters for the design of a door to door orthogonal network. Additionally, a sensitive analysis is representing integrating the change of variables such as demand, city size and cost of infrastructure and electric vehicles. On the other hand, Section 4 indicates the discussion of the results. Finally, general conclusions, Guadalajara achievements and most important remarks are drawn in Section 5.

## 2. Materials and Methods

This section will describe the methodology followed and all the assumptions adopted in the model.

### 2.1. Methodology

It was assumed a rectangular shaped city of area $D_x \cdot D_y$ where the street network resembles a perfect grid layout. The bus network design problem is aimed at minimizing the total cost of the system, composed by the infrastructure and the operating cost incurred by the transit agency, the emission cost caused by vehicles and the time cost by transit users. The modeling approach to design the bus network will be based on the previous work of Daganzo and Estrada et al. [25,40] considering the following features and assumptions:

- Street network and bus corridor structure: It was assumed that buses follow a perfect grid configuration. Another assumption is that the city is covered by orthogonal bus corridors, traveling in the west–east and north–south axis and operated in two directions of service (see Figure 1). The headways of the corridors running along the horizontal and vertical directions ($x$ and $y$ axes) are denoted by $H_x$ and $H_y$ respectively. Moreover, the route spacing between the vertical or horizontal lines are respectively $s_x = p_x \cdot s$ and $s_y = p_y \cdot s$, where $s$ is the stop spacing and $p_x$, $p_y$ two integer variables. Finally, there is a transfer stop at each intersection between vertical and horizontal routes. Hence, users can move in the network from any point to any other point with one transfer.

- User behavior and service characteristics: The network is designed to minimize the total cost at the peak hour, serving a total of $\Lambda$ passengers with origins and destinations uniformly distributed in the service area $D_x$ and $D_y$ (in Estrada [25] it is stated that the agreement between the cost and performance estimations of continuous approximation models with OD simulation techniques is within 85%). However, the bus service will also be provided during $\Omega$ hours along the day, with an average hourly demand $\lambda$(pax/h). The door-to-door travel time of users integrates: the access and egress time spent at a $v_w$ walking speed to/from the closest stop, waiting time

at stops, riding time and the transfer time to overcome the distance $\delta$ between the loading platforms of orthogonal routes.

- Kinematic characteristics of buses: The maximal cruising speed on both horizontal and vertical lines is assumed to be identical and equal to $v$. Boarding and alighting time per passenger at stops is $\tau'$. On the other hand, vehicles have a constant acceleration rate $a$ and a maximal instantaneous speed of $v_{max}$. Therefore, the additional driving time to perform one stop with regard to one vehicle cruising at maximal speed (without stopping) is $\tau = v_{max}/a$.
- Vehicle powertrains and energy provision: It was assumed that ICE-powered buses (diesel) or battery electric buses (BEB) would operate the service. In the case of diesel vehicles, the auxiliary facility to provide energy will be a set of fuel dispensers (fuel station) to be located in the bus garage.

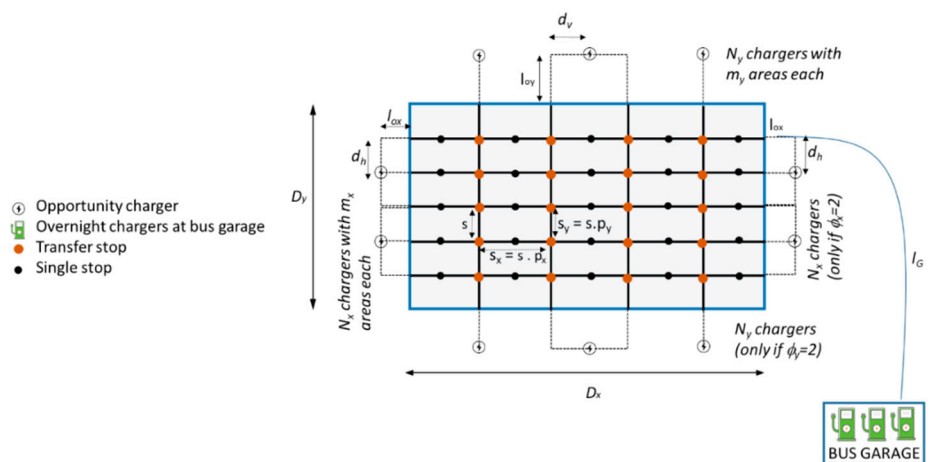

**Figure 1.** Bus network layout defined by spatial and temporal decision variables and charging facilities.

On the other hand, if the service is provided by BEB vehicles, one of the crucial decision variables is the energy capacity of the battery pack ($E$, in kWh). In fact, it was assumed that vehicles must have, at any point of the service, enough energy to run a distance to the bus garage or maintenance location. Despite the fact that the city may present several bus garages scattered in the city periphery, it was assumed that the maximal distance to be run from terminal stop to the bus garage is $l_G$. This can be conceived as a distance margin that every bus may overcome in the necessary case. Assuming an average vehicle consumption factor of $f_c$ (kWh/veh-km), the effective energy that can be used is ($E - f_c \cdot l_G$). Therefore, it has to be check that the effective energy capacity ($E - f_c \cdot l_G$) provides a sufficient autonomy range to operate the service all day long. If so, this vehicle will be charged at the bus garage during the night period. This operation is denoted as overnight charging.

The total number of chargers to be deployed at the bus garage for overnight charging is given by $n_{ch} = M / \left[\frac{24 - \Omega}{E/S_D}\right]^-$, where $M$ is the fleet size (see Equation (12)) and $[x]^-$ a mathematical operator that gives the lower integer of $x$. The term into brackets is, indeed, the number of buses that a single charging station is able to charge completely their batteries at full capacity ($E$) at an electric power $S_D$ during the available night time. Otherwise, if BEB vehicles are equipped with small battery packs, it may be charged on-route. This operation is known as opportunity charging. From now on, it will be assumed that opportunity fast chargers are only located at terminal stops. The regular case considers that there are $N_y$ charging stations evenly distributed along distance $D_x$ in the North or the South periphery quadrant of the city.

Due to urban space scarcity, it was considered that all these stations are located a vertical distance $l_{oy}$ far from the city boundary. Similarly, there are $N_x$ stations located in the east or west periphery quadrant evenly distributed along $D_y$ distance. All these $N_x$

stations are situated $l_{ox}$ distance away from the vertical boundary of the city. Therefore, vertical and horizontal corridors will have one terminal stop in each round trip where the battery charging operation is available. Nevertheless, it can be also considered that $N_x$ and $N_y$ additional charging facilities can also be deployed symmetrically in the opposite periphery quadrants, allowing buses to be charged at each terminal stop of each bus route direction. Doing this, vehicles may be equipped with smaller battery packs since bus charging operation may be more frequent than before. The E-W and N-S periphery quadrants with charging stations are controlled by decision variables $\phi_x$ and $\phi_y$. When these variables are equal to $\phi_x = 2$; $\phi_y = 2$, it means that charging stations are located in both periphery quadrants E-W and N-S.

Depending on the location of the chargers in W-E quadrants, the vehicles operating the last part of a horizontal corridor will run along a vertical detour distance $d_v$ to the charger. This distance is, in fact, a function of the number of chargers located at W-E quadrants, $d_v = d_v(N_x)$. Similarly, vehicles operating vertical corridors will incur an extra horizontal distance $d_h = d_h(N_y)$ based on the number of available chargers. Therefore, when $\phi_x = 2$; $\phi_y = 2$, the distance range between two charging operations in vertical and horizontal corridors are respectively denoted by $d_{c,S} = d_{c,N} = D_y + 2\,l_{oy} + 2d_h$, $d_{c,E} = d_{c,W} = D_x + 2\,l_{ox} + 2d_v$. They include respectively the longitudinal distance run within the city boundaries, the longitudinal distance travelled out of the city boundaries, and the detour distance in the transversal direction.

On the contrary, the situation described by $\phi_x = 1$; $\phi_y = 1$ is equivalent to the regular case, where there is a charging station in just one terminal stop. In that case, the distance run is $d_{c,S} = 2D_y + 2l_{oy} + 2d_h$ and $d_{c,E} = 2D_x + 2l_{ox} + 2d_v$. Hence, the total number of charging stations to provide service in vertical and horizontal corridors is respectively $\theta_y N_y$ and $\theta_x N_x$. Note that when the charging time of a single bus at a terminal stop is higher than the target time headway $H_x$ or $H_y$, vehicles cannot be dispatched at the required frequency unless multiple charging areas are deployed. Therefore, several buses must be charged simultaneously at the terminal stop. Hence, it was considered that each charging station $y$ ($y = 1, \ldots, N_y$) located in the N–S quadrants will be deployed with $m_y$ charging areas. Each charging area has a parking slot for one bus and is equipped with an independent charging connector. Therefore, a total of $m_y$ buses can be charged simultaneously at each charging station of the set $N_y$. Similarly, each charging station $x$ ($x = 1, \ldots, N_x$) located in E-W periphery quadrants will be equipped with $m_x$ charging areas.

The number of charging areas in each charger station ($m_x$ or $m_y$) is defined to be the minimal value that satisfies the required charging time and the dispatchment of vehicles at the target headways $H_x$ or $H_y$. To do so, it was considered that the charging time of one vehicle at charging station $y$ is $T_{c,y}$ ($y = 1, \ldots, N_y$). In addition to that, it was considered that the expected number of routes served at each charging station is $\frac{D_x}{N_y p_x s}$. If the average headway between two consecutive bus departures from each charger $y$ is $\overline{h}_y$, therefore the minimal number of charging areas that each terminal stop $y$ in north and/or south peripheral quadrants must present is given by $m_y = \left[\frac{T_{c,y}}{\overline{h}_y}\right]^{+} = \left[\frac{T_{c,y}}{H_y}\frac{D_x}{N_y p_x s}\right]^{+}$. In the W-E periphery quadrants, the number of charging areas is similarly calculated by $m_x = \left[\frac{T_{c,x}}{H_x}\frac{D_y}{N_x p_y s}\right]^{+}$. Therefore, the total number of opportunity charging areas is $n_{ch} = (m_x\phi_x N_x + m_y\phi_y N_y)$.

Another available charging alternative is the daily charging operation at the bus garage, where vehicles are charged at this facility during the service. As a difference from the opportunity and overnight charging, vehicles present very different levels of SOC during the service. The modeling approach in the following lines is developed for a horizontal route. Let $\mu_x = 2D_x/(H_x \cdot v_{cx})$ be the number of vehicles needed to operate a single horizontal corridor to fulfill the target headway $H_x$. It was assumed that after $\Delta t_0$ units of time from the beginning of the service, a subset of $\Delta\mu_x$ vehicles ($\Delta\mu_x \leq \mu_x$) will go to the bus garage to be charged. These vehicles will be replaced in

the service by the same amount of extra vehicles that were previously parked at this facility, with batteries completely charged. These $\Delta\mu_x$ vehicles will spend $l_G/v$ units of time traveling to the bus garage, $\Delta t_{0,x} \cdot v_{c,x} \cdot \frac{f_c}{S_d}$ units of time being charged (they have only provided service during $\Delta t_{0,x}$), and $l_G/v$ units of time for traveling again to the bus route to continue the service. Therefore, the time needed during this first cycle of charging is $\Delta t_{1,x} = \frac{2l_G}{v} + \Delta t_{0,x} \cdot v_{c,x} \cdot f_c / S_d$. At time $\Delta t_{0,x} + \Delta t_{1,x}$, we can start the second cycle of charging, replacing those $\Delta\mu_x$ vehicles pending to be recharged, by the ones coming from the bus garage. Note that the charging operation of these second group of vehicles needs $\Delta t_{2,x} = \frac{2l_G}{v} + (\Delta t_{0,x} + \Delta t_{1,x}) \cdot v_{c,x} \cdot f_c / S_d$ units of time to be reintroduced again into the service. Generally, the $\Delta\mu_x$ vehicles replaced at the $n$-th cycle of charging will consume $\Delta t_{n,x} = \frac{2l_G}{v} + \left(\sum_{i=0}^{n-1} \Delta t_{i,x}\right) \cdot v_{c,x} \cdot \frac{f_c}{S_d}$ units of time for being available again. Therefore, this charging scheme may charge vehicles during the whole service at the expense of deploying $\Delta\mu_x$ additional vehicles in the operation, and performing the unproductive trips between routes and bus garage.

The variables that control the charging operations are the number of charging cycles performed ($\pi_x$), the number of extra vehicles ($\Delta\mu_x$) and the elapsed time when the first charging operation is performed ($\Delta t_{0,x}$). In order to ensure a feasible charging scheme, it has to fulfill on one hand that all charging cycles are performed within the service period ($\sum_{i=0}^{\pi_x} \Delta t_{i,x} \leq \Omega$) and, secondly, the added vehicles and the number of charging cycles are enough to charge all vehicles operating the route ($\pi_x \cdot \Delta\mu_x \geq \mu_x$). Similar equations must be fulfilled in the vertical corridors ($y$ coordinate). Consequently, the bus network design problem is aimed at minimizing the total costs of the system (Z), defined by Equation (1).

$$\min_{\substack{s,\ H_x, H_y, px, py, \\ Nx,\ Ny, \phi_x,\ \phi_y}} \{Z = [c_L L + c_N n_{ch}] + [c_V V + c_M M + c_B E \times M] + [\lambda \beta_T T] + [Z_E]\} \tag{1}$$

subject to

$$O_x \leq C \tag{2a}$$

$$O_y \leq C \tag{2b}$$

$$\max(v_{c,x}, v_{c,y}) \times \Omega + l_G \leq \frac{E}{f_C} \tag{3a}$$

$$\max\left(\sum_{i=0}^{\pi_x-1} \Delta t_{i,x} \times v_{c,x}, \sum_{i=1}^{\pi_x} \Delta t_{i,x} \times v_{c,x}, \sum_{i=0}^{\pi_y-1} \Delta t_{i,y} \times v_{c,y}, \sum_{i=1}^{\pi_y} \Delta t_{i,y} \times v_{c,y}\right) \leq \frac{E}{f_C} \tag{3b}$$

$$d_{c,W} + l_G, d_{c,E} + l_G, d_{c,N} + l_G, d_{c,S} + l_G < \frac{E}{f_C} \tag{3c}$$

$$s \geq s_{min};\ H_x, H_y \geq H_{min} \tag{4}$$

$$s, H_x, H_y \in \mathbb{R}^+ \tag{5}$$

$$N_x, N_y \in \mathbb{N} \tag{6}$$

$$p_x, p_y, \phi_x, \phi_y = \{1, 2\} \tag{7}$$

The first term in brackets in Equation (1) captures the infrastructure cost that the transit agency will incur, where $L$ is the total length of bus corridors (two ways of service) and $n_{ch}$ the number of electric charging areas (BEB vehicles) or the number of refueling stations (ICE vehicles). The corresponding parameters $c_L$ and $c_N$ determine the depreciation cost per unit of bus lane distance, and per unit of charging area and hour respectively. In the case of diesel vehicles, $c_N$ is calculated as the depreciation cost of one fuel station per hour, divided by the number of refueling operations that a single station can perform along the night period.

The second term in brackets estimates the agency's operating cost. The total distance run by the fleet in one hour ($V$) and the fleet needed ($M$) are multiplied by the corresponding

unit distance cost ($c_V$) and unit temporal cost ($c_M$). The former considers all operating costs depending on the distance run (mainly energy), while $c_M$ embraces vehicle depreciation, driver salary, insurances and other fixed expenses. Typically, the depreciation cost of electric buses does not encompass the depreciation cost of batteries ($c_B$), since the battery lifetime is significantly lower than the vehicles. Indeed, the battery capacity ($E$) has to be determined, according to the route length and energy consumption factors.

Moreover, the user cost contribution is calculated in the third bracketed term by the product of the average hourly demand rate ($\lambda$ in pax/h), the value of time ($\beta_T$ in USD/pax-h) and the total door-to-door travel time ($T$, in hours). This travel time component encompasses access ($A$), waiting ($W$), transfer ($Tr$) and in-vehicle travel time ($IVTT$). Finally, it was also included in the objective function the monetary cost of emissions per hour of service ($Z_E$). Therefore, the total cost function $Z$ (USD/h), in terms of monetary units per hour of service.

Equations (2a) and (2b) constraint that the occupancy of vehicles running along $x$ and $y$ directions ($O_x$ and $O_y$ respectively) should be lower than the maximal passenger capacity of buses ($C$ in pax/bus). The set of Equation (3) is aimed at verifying that vehicles present a sufficient range under three charging schemes. If we opted for an overnight charging scheme at the bus garage, Equation (3a) states that the total distance run along the whole day by BEB vehicles, plus a safety margin distance $l_G$, should be lower than the maximal allowable distance ($E/f_C$). In the former equation, $v_{c,x}$ and $v_{c,y}$ represent the number of kilometers run per hour in the $x$ and $y$ direction by a single bus (net commercial speed calculated through Equations (11a) and (11b). Equation (3b) ensures a sufficient range for the group of vehicles charged at the first ($i = 1$) and last cycle ($i = \pi_x$ or $\pi_y$) of charging at the bus garage during the service. Similarly, Equation (3c) should be only satisfied by BEB technology with the opportunity charging scheme. Equation (4) constraints the minimal threshold of stop spacing ($s_{min}$) and headways ($H_{min}$). It was assumed that below these two thresholds, buses would stop too frequently along the route or they would not be able to maintain an acceptable headway adherence. Finally, constraints (5)–(7) specify the feasible domains and the nature of the decision variables. Indeed, it was restricted that routes will be spaced, at most, twice the stop distance.

In the following subsections, it will calculate each component of the objective function dependent on the decision variables of the problem: s, $H_x$, $H_y$, $p_x$, $p_y$, $N_x$, $N_y$, $\phi_x$ and $\phi_y$.

**Bus agency metrics**

The different cost components incurred by transit agencies are estimated, considering similar approaches presented in Daganzo and Estrada [17,25]. Equation (8) captures the total bidirectional length of the network ($L$), while Equation (9) estimates the distance run in one hour by the whole fleet ($V$). It was assumed that in the case of BEB vehicles with opportunity charging scheme, each vehicle operating a horizontal corridor runs an extra distance $l_{ox} + d_v$ from the terminal stop to the closest charging station. This extra distance is equal to $l_{oy} + d_h$ for vehicles running along vertical corridors. In diesel and BEB powertrains, $l_{ox} = d_v = l_{oy} = d_h = 0$. In the case of day charging at the bus garage, the terms $\sum_{i=0}^{\pi_x} \Delta t_{i,x}$ and $\sum_{i=0}^{\pi_y} \Delta t_{i,y}$ represent the time needed to charge all vehicles starting the service (one charging cycle) in the horizontal and vertical corridors. In other charging schemes and diesel powertrains, these terms must be $\sum_{i=0}^{\pi_x} \Delta t_{i,x} = \sum_{i=0}^{\pi_x} \Delta t_{i,x} = \infty$. Moreover, it was calculated the expected number of transfers $p(1)$ per user through Equation (10). This variable is needed to compute the net commercial speed $v_c$ in $x$ and $y$ coordinates, addressed in Equation (11). The net commercial speed is the average speed that buses will experience to perform a complete round trip of length $2D_x$ (horizontal) or $2D_y$ (vertical). In the case of BEB with opportunity charging scheme, the total length is computed by $2(D_x + l_{ox} + d_v)$ (horizontal) or $2(D_y + l_{oy} + d_h)$ (vertical). The net commercial speed takes into account the lay-over time at terminal stops ($\theta_E, \theta_W, \theta_S, \theta_N$) and the charging time

$T_{c,z}$ spent at the corresponding terminal station $z$ ($z$ = $W$, $E$, $S$ or $N$) only by BEB vehicles. Finally, the fleet size needed to run the service is addressed by means of Equation (12).

$$L = D_y D_x \left( \frac{1}{p_y s} + \frac{1}{p_{xy} s} \right) \tag{8}$$

$$V = 2 D_y D_x \left( \frac{1}{H_x p_y s} + \frac{1}{H_y p_x s} \right) + \frac{2 D_y}{H_x p_y s} \left( l_{ox} + d_v + \frac{2 l_G D_x}{v_{c,x} \sum_{i=0}^{\frac{\pi_x}{s}} \Delta t_{i,x}} \right)$$
$$+ \frac{2 D_x}{H_y p_x s} \left( l_{oy} + d_h + \frac{2 l_G D_y}{v_{c,y} \sum_{i=0}^{\frac{\pi_y}{s}} \Delta t_{i,y}} \right) \tag{9}$$

$$P(1) = 1 - \frac{p_x s D_y + p_y s D_x - p_x p_y s^2}{D_x D_y} \tag{10}$$

$$\frac{1}{v_{c,x}} = \frac{1}{v} + \frac{1}{s}\tau + \frac{\frac{\Lambda}{2} H_x [1 + p(1)] \tau'}{2 D_y (D_x + l_{ox} + d_v)/(p_y s)}$$
$$+ \frac{\theta_E + \theta_W + \max\{0, T_{c,E} - \theta_E\} + \max\{0, T_{c,W} - \theta_W\}}{D_x + l_{ox} + d_v} \tag{11a}$$

$$\frac{1}{v_{c,y}} = \frac{1}{v} + \frac{1}{s}\tau + \frac{\frac{\Lambda}{2} H_y [1 + p(1)] \tau'}{2 D_x (D_y + l_{oy} + d_h)/(p_x s)}$$
$$+ \frac{\theta_S + \theta_N + \max\{0, T_{c,S} - \theta_S\} + \max\{0, T_{c,N} - \theta_N\}}{D_y + l_{oy} + d_h} \tag{11b}$$

$$M = \frac{2 D_y}{(v_c)_x H_x p_y s}(D_x + l_{ox} + d_v) + \frac{2 D_x}{(v_c)_y H_y p_x s}(D_y + l_{oy} + d_h) + \Delta \mu_x \frac{D_y}{p_y s}$$
$$+ \Delta \mu_y \frac{D_x}{p_x s} \tag{12}$$

In the case of BEB with the opportunity charging scheme, the charging time $T_{c,z}$ spent at terminal station $z$ is calculated in Equation (13), where $z$ represents the charging stations located on the north, south, east and west part of the city. The first term is the minimal charging time whose estimation is based on the energy consumed in the distance run from the last charging operation $d_{c,z}$, the energy consumption factor $f_c$ and the charging speed $S_D$. The second term captures the positioning and clearance time spent at the charging area per bus in the terminal stop, $t_{pos}$. This time is neglected in alternative solutions that do not visit the on-street opportunity charging facility ($T_{c,z}$ = 0).

$$T_{c,z} = \frac{d_{c,z} f_c}{S_D} + t_{pos} \tag{13}$$

**User metrics**

The expected door-to-door travel of a single user $T$ is calculated as the summation of access ($A$), waiting ($W$), transfer ($Tr$) and in-vehicle travel time ($IVTT$) through Equations (14)–(17) respectively. Equation (15) the estimation of the waiting time specifies the different headway provision along directions $x$ and $y$. Moreover, the $IVTT$ term must take into account the gross commercial speed of buses $v'_c$, i.e., the average commercial speed experienced by users, neglecting the idle time spent at the terminal stops. Finally, Equation (18) defines the expected passenger occupancy in the vehicles operating routes in the $x$ and $y$ directions.

$$A = s \left( \frac{2 + p_x + p_y}{2 v_w} \right) \tag{14}$$

$$W = (1 - p(1)) \times \frac{H_x + H_y}{4} + p(1) \frac{H_x + H_y}{2} \tag{15}$$

$$T_{TR} = \frac{\delta \cdot p(1)}{v_w} \tag{16}$$

$$IVTT = \frac{D_x + D_y}{3} \left( \frac{1}{v} + \frac{1}{s}\tau \right) + \frac{\Lambda s [1 + p(1)] \tau' (D_x H_x p_y + D_y H_y p_x)}{12 D_y D_x} \tag{17}$$

$$O_x = \frac{\Lambda}{16}(1 + p(1))\frac{s_y}{D_y}H_x \quad O_y = \frac{\Lambda}{16}(1 + p(1))\frac{s_x}{D_x}H_y \tag{18}$$

**Emissions**

The estimation of the local emissions and greenhouse gases (GHGs) generated within all the processes needed to run the bus service is addressed in this section. These emissions have been calculated by means of a lifecycle analysis of the vehicle manufacturing, infrastructure construction and the well to wheel analysis of the energy required to operate the vehicles, given a bus powertrain and bus size. These contributions are estimated by means of the corresponding emission factor of a specific pollutant generated by one process (vehicle operation, manufacturing and infrastructure construction). These factors will be later multiplied by the corresponding size fleet ($M$), infrastructures ($L$ and $n_{ch}$) and distance run ($V$) in one hour of service.

The local pollutants considered are $PM_{10}$, $SO_x$, $CO$, $NO_x$, $NH_3$ and VOC, while $CO_2$, $CH_4$ and $N_2O$ are transformed into $CO_2$ eq to account for GHG, taking the corresponding warming potentials. Both local pollutants and $CO_2$ eq constitute the set $P$ of different emission typologies considered in this paper. Finally, the mass of each pollutant generated per hour of service has been monetized to be integrated in the objective function. To do so, a proxy parameter $\varepsilon_p$ has been defined to monetize the effect of producing one unit of pollutant $p \in P$. The values of these proxy parameters (in USD/g of pollutant $p$) were determined according to van Essen et al. [40]. The processes considered that have an associated economic impact in terms of emissions and pollutants in the bus system are the following:

Monetization of the emissions caused by the circulation of vehicles: $Z_{e,F}$: Each ICE vehicle produces $E_{F,p}$ (g/veh-km) grams of local pollutant $p$ ($p \in P$) per kilometer run. In fact, these emission factors capture the pollutants generated in the tank-to-wheel phase. These emission factors, in reality, depend on the type of vehicle, speed regime, driving style and fuel consumption factors [33]. The emission monetization associated to the circulation of vehicles in one hour can be estimated by Equation (19), taking into account the total distance run by the whole fleet ($V$) in the peak hour. It is considered that BEBs do not present any pollutant contribution.

$$Z_{e,F} = V \times \sum_{p \in P} E_{F,p}\varepsilon_p \tag{19}$$

Monetization of the emissions caused by the energy generation in production plans: $Z_{e,E}$: The generation of electricity from primary energy sources for bus battery charging also lead to emissions when the service is operated by BEBs. These emissions, in both BEB and ICE vehicles, are estimated by means of the well-to-tank analysis. The emission factor $E_{E,p}$ of pollutant $p$ (g of pollutant $p$/kWh) is estimated as the mass of this pollutant caused in the generation and transportation of one unit of energy to move vehicles. These emission factors depend on the energy mix of the country. Equation (20) captures the estimation of the $Z_{e,E}$ variable, where $f_c$ (kWh/veh-km) represents the energy consumption factor of the bus fleet.

$$Z_{e,E} = V \times f_c \sum_{p \in P} E_{E,p}\varepsilon_p \tag{20}$$

Monetization of the emissions caused by vehicle manufacturing and maintenance: $Z_{e,M}$: The manufacturing of public transport vehicles consumes and transforms a wide number of raw and semiprocessed materials. The emission factor of pollutant $p$, $E_{M,p}$ (g of pollutant $p$/veh-h) is defined as the ratio among the total amount of emissions of this pollutant $p$ in the vehicle production and maintenance and the number of working hours along the vehicle lifetime. The emission monetization of this effect is obtained by Equation (21), multiplying emission factors by the proxy parameters ($\varepsilon_p$) and the fleet size ($M$).

$$Z_{e,M} = M \times \sum_{p \in P} E_{M,p}\varepsilon_p \tag{21}$$

Monetization of the emissions caused by the row infrastructure construction: $Z_{e,I}$: The manufacturing of the infrastructure needed to build that transit lane also contribute to the GHG and local emissions. In that case, the emission factor for pollutant $p$ is considered per unit of infrastructure length and unit of time used, $E_{I,p}$ (g/km-h). It was also accounted for the contribution of public work machinery used in the whole construction phase of the bus network. The total amount of emissions produced in one kilometer is split along the target lifetime of the infrastructure. Then, the monetization can be obtained by Equation (22).

$$Z_{e,I} = L \times \sum_{p \in P} E_{I,p}\, \varepsilon_p \tag{22}$$

Monetization of the emissions caused by the stop/station construction: $Z_{e,S}$: This term encompasses the economic effects of the emissions generated in the construction of stops, stations and charging stations. Traditionally, the material and energy consumption in the construction of the bus stop was significantly lower than the bus lane infrastructure. It was assumed that $E_{S,p}$ (g/st-h) and $E_{C,p}$ (g/st-h) are the emission factors of pollutant $p$ per hour of stop and electric charging area respectively. The emissions caused by the construction of refueling stations for diesel vehicles at bus garage are considered in the $E_{E,p}$ term, and therefore they were not addressed here. Consequently, the term $Z_{e,S}$ can be calculated by Equation (23).

$$Z_{e,S} = \frac{D_x D_y}{p_x p_y s^2} \times \sum_{p \in P} E_{S,p}\, \varepsilon_p + n_{ch} \times \sum_{p \in P} E_{C,p}\, \varepsilon_p \tag{23}$$

**Optimization procedure**

The analytical model presented before allows identifying the optimal design of the public transport network from 5 decision variables for ICE powertrains, or for BEB powertrain when overnight charging is chosen: headways in horizontal and vertical routes, stop spacing and route spacing between vertical and horizontal lines defined by the auxiliary variables $p_x$ and $p_y$. Nevertheless, in the case of the deployment of electric bus technologies with opportunity charging, four additional decision variables are considered to account for the number of electric charging stations and charging areas in the N–S periphery ($\theta_y$, $N_y$) or in the E-W periphery ($\theta_x$, $N_x$). The optimization process followed in this paper was based on a grid enumeration of the objective function, in relation to the decision variables. For each of the variables with a continuous domain, it was identified an enumeration interval pace ($\Delta s$ and $\Delta H$) to discretize the enumeration domain. The discretized domain of these variables is therefore enumerated by $s = s_{min} + k_s \Delta_s$ and $H_x = H_{min} + k_H \Delta_H$, $H_y = H_{min} + k_V \Delta_H$; $k_s, k_H, k_V = 0, .., k_{max}$. Input parameters $s_{min}$, $H_{min}$ define the minimal reasonable value of stop spacing and time headway respectively, while $k_{max}$ is defined based on the maximal value of the former decision variables. In the case of the discrete variables, the enumeration domain of $p_x$, $p_y$ and $\phi_x$, $\phi_y$ was {1, 2}, while the corresponding to the number of chargers was $N_x = \{1, 2, \ldots, \frac{D_y}{s \cdot p_x}\}$ and $N_y = \{1, 2, \ldots, \frac{D_x}{s \cdot p_y}\}$. Therefore, the total cost of the collective transport system $Z(s', H', p_x', p_y', N_x', N_y', \phi_x', \phi_y')$ is evaluated by Equation (1) for each potential combination of decision variables defined in the grid search. Hence, the feasible solution of the optimization problem stated in Equation (1) is the combination of decision variable values that produce the minimal cost of the objective function ($Z$), satisfying the constraints of Equations (2)–(7).

*2.2. Materials*

The applicability of the model presented has been tested under various scenarios of bus powertrains in Guadalajara, México (Figure 2). The city of Guadalajara, located in the Jalisco state, is the second most populated city in Mexico, with near to 5 million inhabitants within its metropolitan area spread in an area of 151.6 km² [41].

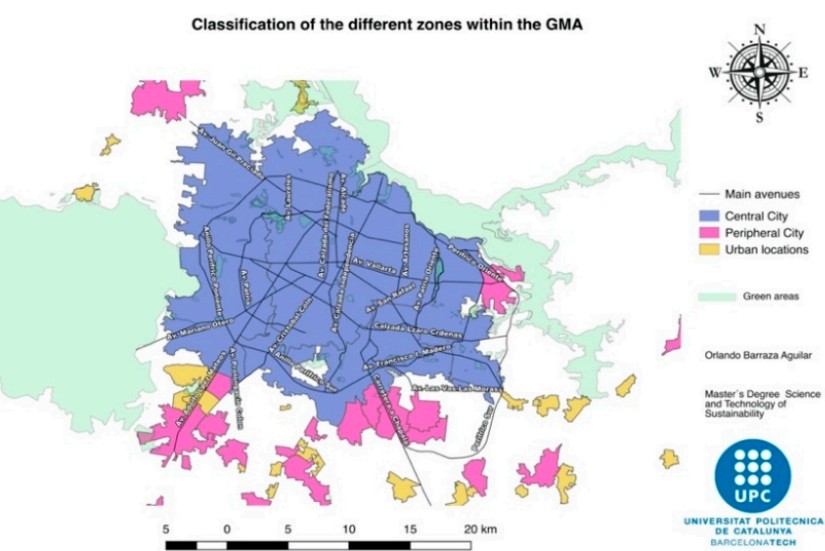

**Figure 2.** Indicates the Guadalajara Metropolitan Area with its corresponding urban division.

Public transport services the Guadalajara Metropolitan Area by means of a door-to-door bus network, operated by 12 m-long diesel buses. The bus service area has been modeled by a rectangle of sides $D_x \times D_y$ = 18·15 km². The peak hour demand of bus passengers in a working day was estimated to be $\Lambda$ = 333,613 pax/h, while the average hourly demand considering the variability along the whole day was $\lambda$ = 236,605 pax/h. Both figures have been calculated from the data provided in Parada [42]. Moreover, it was assumed that the value of time of passengers using the bus service in Guadalajara is $\mu$ = 3.14 USD/pax-h, considering that the average monthly salary in 2018 was 514 USD [43]. It was assumed that these users show an average walking speed of $v_w$ = 4.5 km/h and the average transfer distance between loading areas at stops is $\delta$ = 0.3 km [25]. The service is provided along $\Omega$ = 16 h per day.

The infrastructure cost to build the bus lane along the transit corridors was supposed to be $c_L$ = 84.36 USD/km-h [23]. Eight scenarios regarding different vehicle capacities, powertrains and charging operation are considered in the analysis:

- C-12. Standard bus with diesel engine (12 m long, conventional environmental label);
- C-18. Articulated bus with diesel engine (18 m long, conventional environmental label);
- EuroVI-12. Standard bus with diesel engine (12 m long, Euro VI environmental label);
- EuroVI-18. Articulated bus with diesel engine (18 m long, Euro VI environmental label);
- BEB-12 Ov. Standard battery-electric bus with overnight charging (12 m long);
- BEB-12 Opp. Standard battery-electric bus with opportunity charging (12 m long);
- BEB-12 Day. Standard battery-electric bus with charging at the bus garage during service (12 m long);
- BEB-18. Articulated battery-electric bus with opportunity charging (18 m long).

The scenario C-12 represents the current standard size (12 m long) and diesel vehicle powertrain use in Guadalajara, without significant treatments of emissions (conventional diesel). A similar scenario entitled C-18 was created for articulated diesel buses (18 m long) with the same conventional environmental label but higher passenger capacity. Scenarios EuroVI-12 and EuroVI-18 will consider a situation with the most recent diesel-hybrid technology. They differ in terms of the vehicle typology, and consequently, in the maximal passenger occupancy that they can accommodate. Finally, the last four scenarios suppose that charging facilities and BEB vehicles would be deployed. For standard buses of 12 m of length, it was considered that overnight charging at the bus garage, day charging at the bus garage during service and opportunity charging at terminal stops are feasible.

For articulated buses of 18 m, only the opportunity charging operation was considered in the analysis.

The vehicle capacities of all standard and articulated buses are respectively $C$ = 70 pax and $C$ = 120 pax. It was assumed that buses of all the scenarios present the same cruising speed $v$ = 30 km/h, unit boarding and alighting time per passenger $\tau'$ = 3 s [18], and the additional running time to perform one stop due to acceleration and deceleration phase is $\tau = \frac{v}{a}$ = 35 s. On the other hand, the maximal acceleration/deceleration rate is $a$ = 0.85 m/s$^2$ [44]. The parameters that depend on the vehicle typology are summarized in Table 1. The driver cost component (Table 1, row c) was reported to be 9.45 USD/veh-h in Guadalajara, considering the current salaries [45]. The unit distance cost of the ICE vehicles in the first four scenarios was calculated considering the cost of the diesel consumption [32], and an additional cost term that accounts for vehicle maintenance staff and spare parts.

The latter cost component (row b), refuel workload (row e) and engineering staff (row f in Table 1) were calculated considering the real cost reported in Barcelona by Transportes Metropolitanos de Barcelona (TMB) [46] and multiplied by the ratio $\eta_{GB}$ = 0.2505. This ratio, calculated as the driver cost in Guadalajara divided by the corresponding value in Barcelona (37.72 Euro/veh-h), accounts for the different labor regulations, salaries and monetary parity between these cities. The vehicle purchasing costs, charger and refueling station investments are supposed to be equivalent to the figures reported in TMB [46]. The corresponding cost per hour of vehicle or hour of facility were calculated through a linear amortization along 79,200 h (15 years·330 day/year·16 h/day). The cost of each refueling station is divided by the number of buses whose fuel tank can be refilled in the same day (350 buses), and expressed in terms of USD/veh-h. It is worth to mention that the investment in the charging stations for standard buses at the bus garage is significantly lower than the corresponding to buses charged on the street (opportunity charging).

The battery cost presents a significant variability depending on the different available chemistries, lifespans and acquisition conditions. Organization for Economic Co-operation and Development and International Energy Agency (OCDE-IEA) [45] suggests that the battery cost should range between 120 and 400 USD/kWh in all BEB analysis. Indeed, they suggest a normal value of 260 USD/kWh. Nevertheless, in our study, the resulting battery cost was calculated as a straight-line depreciation of the maximal purchasing cost threshold 400 USD/kWh, along a lifetime of 4 years.

The unit emission costs ($\varepsilon_p$) and emission factors ($E_{F,p}$, $E_{E,p}$, $E_{M,p}$, $E_{I,p}$, $E_{C,p}$) for each pollutant $p \in P$, effect and vehicle scenario in the case applicability in Guadalajara are presented in detail in the Table A1 in the Appendix B. The unit emission costs parameters were considered the average values of contingency cost proposed by van Essen et al. [40] for the 28 European Union countries. In the case of $CO_2$ eq, the monetization factor was considered $\varepsilon_{CO2} = 111 \; USD/Ton \; CO_2$ based on the impact and wears on the territory. Reasonably, it is one order of magnitude greater than the figure proposed in SENDECO, since the latter is based on the market equilibrium price of $CO_2$ emission rights among countries.

The terms $E_{F,p}$ accounting for the tank-to-wheel emissions are calculated considering EEA using the Tier 3 estimation for the urban buses standard 15–18 t and urban buses articulated >18 t (diesel conventional and diesel EURO VI diesel powertrains), when the road slope is 0 and it is fully loaded. The non-exhaust emissions like fuel evaporation or emissions generated in the brake operations are not considered. Therefore, the corresponding emissions factors of this category for electric buses are considered to be 0. The emissions $E_{E,p}$ produced in the energy generation and transportation for diesel and battery electric bus powertrains within the well-to-tank phase were calculated using the GREET© fuel cycle model v2019 [30]. It considered the electricity mix for Mexico in 2014 published in IEA [47] based on the following primary energy sources: coal 11.3%, oil 11%, gas 57.1%, nuclear 3.3%, hydro 13%, bioenergy 0.3% and others clean sources (4%).

The emissions $E_{M,p}$ caused by the vehicle manufacturing process are approximated to the results presented in Nordelöf et al. [35] for standard diesel and electric buses, based on the data of Volvo's 7900 model series. The corresponding values for articulated buses were

supposed to be 1.35 times higher than the values for standard buses. The emissions were split by a total vehicle lifetime of 15 years (diesel and electric), assuming 330 days of work along the year and 16 working hours per day.

**Table 1.** Input parameters.

| Cost Parameters | | Standard Bus, 12 m Long | | | | Articulated Bus, 18 m Long | | |
|---|---|---|---|---|---|---|---|---|
| | | Diesel Diesel | | Electric | | Diesel Diesel | | Electric |
| | | Conventional | EURO VI | Over-n./Day | Oppor-tunity | Conventional | EURO VI | Oppor-tunity |
| Energy consumption factor (kWh/veh-km) | | 6.207 | 4.746 | 1.400 | | 7.689 | 6.304 | 1.900 |
| Unit energy cost (USD/veh-km) | (a) | 0.685 | 0.524 | 0.112 | | 0.849 | 0.696 | 0.152 |
| Spare parts, maintenance staff cost (USD/veh-km) | (b) | 0.254 | 0.254 | 0.169 | | 0.293 | 0.293 | 0.163 |
| **Unit distance cost, $c_V$(USD/veh-km)** | **(a + b)** | **0.940** | **0.778** | **0.281** | | **1.142** | **0.989** | **0.315** |
| Unit driver cost, (USD/veh-h) | (c) | 9.450 | 9.450 | 9.450 | | 9.450 | 9.450 | 9.450 |
| Vehicle acquisition cost (USD/veh) | | 198,000 | 277,500 | 555,000 | | 275,000 | 385,000 | 876,900 |
| Amortized Vehicle cost (USD/veh-h) | (d) | 2.500 | 3.504 | 7.008 | | 3.472 | 4.861 | 11.072 |
| Refuel workload at bus garage (USD/veh-h) | (e) | 0.183 | 0.083 | 0.000 | | 0.183 | 0.183 | 0.000 |
| Insurances, control, engineering staff (USD/veh-h) | (f) | 2.626 | 2.626 | 2.964 | | 2.626 | 2.626 | 2.626 |
| **Unit temporal cost, $c_M$ (USD/veh-h)** | **(c + d + e + f)** | **14.759** | **15.663** | **19.421** | | **15.731** | **17.120** | **23.148** |
| Charger facility/Fuel station invest.(kUSD) | | 5550 | 5550 | 95.87 | 777 | 5550 | 5550 | 777 |
| Vehicle capacity of the charger/station (veh) | (g) | 350 | 350 | 1 | 1 | 350 | 350 | 1 |
| Facility cost (USD/charger-h) | (h) | 35.04 | 35.04 | 0.605 | 5.396 | 35.04 | 35.04 | 5.396 |
| Charger/Refuel station maintenance cost (USD/facility-h) | (i) | 1.354 | 1.354 | 0.625 | 0.625 | 1.354 | 1.354 | 0.625 |
| **Unit energy facility cost or refueling station, $c_N$ (USD/charger-h)** | **(h + i)/(g)** | **0.104** | **0.104** | **1.230** | **6.021** | **0.104** | **0.104** | **6.021** |
| Unit battery cost, $c_b$ (USD/kWh-h) | | – | – | 0.0190 | | – | – | 0.019 |
| Aggregated Tank to Wheel emission monetization, $\sum_{p \in P} E_{F,x} \varepsilon_p$(USD/veh-km) | | 0.9015 | 0.1580 | 0 | | 1.1434 | 0.2027 | 0 |
| Aggregated Well to Tank emission monetization, $\sum_{p \in P} E_{E,p} \varepsilon_p$(USD/kWh) | | 0.0106 | 0.0106 | 0.0974 | | 0.0106 | 0.0106 | 0.0974 |
| Aggregated Vehicle manuf. emission monetization, $\sum_{p \in P} E_{M,p} \varepsilon_p$(USD/veh-h) | | 0.1598 | 0.1598 | 0.2008 | | 0.2157 | 0.2157 | 0.2711 |
| Aggregated Infrastructure emission monetization, $\sum_{p \in P} E_{I,p} \varepsilon_p$(USD/km-h) | | 0.1089 | 0.1089 | 0.1089 | | 0.1089 | 0.1089 | 0.1089 |
| Aggregated Charger emission monetization, $\sum_{p \in P} E_{C,p} \varepsilon_p$(USD/char-h) | | – | – | 0.0171 | | | | 0.0171 |

On the other hand, the emissions $E_{I,p}$ caused by the infrastructure construction were calculated based on Wayman et al. [48] to build 1 square meter of pavement surface. Assuming a lifespan of 60 years (including maintenance works, 360 days·16 h of service/day) and a bus lane width of 3.5 m, the $CO_2$ and $SO_2$ emissions were calculated per hour and kilometer of bus lane. Emissions factors of other pollutants were not available. Finally, the emissions factors associated to each stop construction were negligible in our analysis due to the minor requirements of bus shelters in Guadalajara.

Nevertheless, the emission factors for charging facilities were calculated based on Nansai et al. [49] regarding the installation and manufacturing phase of one charger (14.55 Ton $CO_2$, 16 kg $NO_x$, 15 kg $SO_x$ and 13 kg CO). Additionally, it was also considered the emissions of the charger transportation along 40 km from the assembling plant to the final location with a heavy diesel duty truck 14–20 t (EURO VI). For each effect, the summation for all pollutants *p* of the corresponding emission factor, times the unit emission cost is summarized in Table A1. In the case of pollutant $x = CO_2$ eq, the emission factors

considering the vehicle manufacturing, operation and the well to tank phase in Scenarios C-12, Euro VI 12 and BEB 12 Ov. are 3153, 2763 and 2306 g $CO_2$/veh-km (see Appendix B). In the case of the electric vehicle (Scenario 5), the well to tank emission factor is 514 g/kWh, similar to the 582 g/kWh for México [50].

## 3. Results

This section shows the final results derived from the model. Figure 3 shows the total cost of the bus system obtained by the optimization model in different powertrain scenarios, and in the current Guadalajara bus network configuration, operated by conventional standard (12 m) diesel buses.

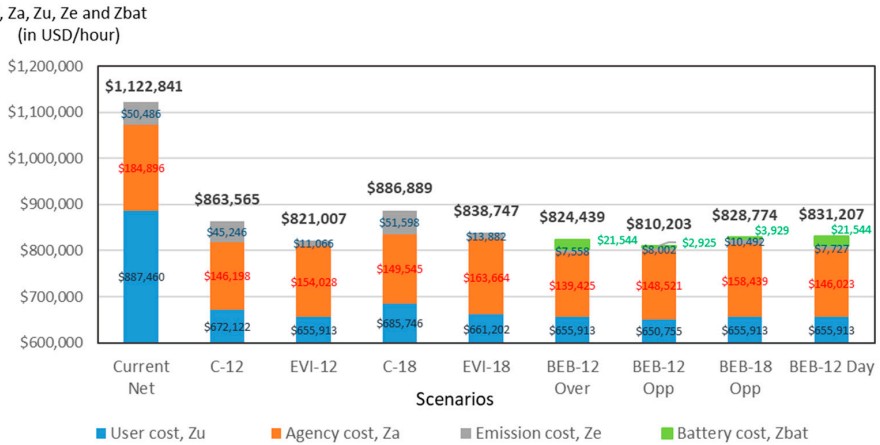

**Figure 3.** Total cost and cost components incurred in the provision of the bus service in the peak hour.

Figure 3 indicates the total cost of each powertrain scenario expressed in USD/hour of service. Total cost integrates: user cost, agency cost, emission cost ad in the electric cases battery cost. On the other hand, Figure 4 summarized the design variables ($p_x$, $p_y$, $s$, $H_x$ and $H_y$) and more performance metrics for each scenario.

Figure 4 is divided in two sections. The section above shows the decision variables, which are $p_x$, $p_y$, $s$, $H_y$ and $H_x$. Each group of lectures corresponds to each of the powertrain scenario proposed. On the other hand the below section indicates the performance metrics of each scenario taking some of the main metrics such as the size of the fleet ($M$), waiting time ($W$), the total distance run by the fleet in one hour ($V$), longitude of the network ($L$) and in-vehicle travel time ($IVTT$).

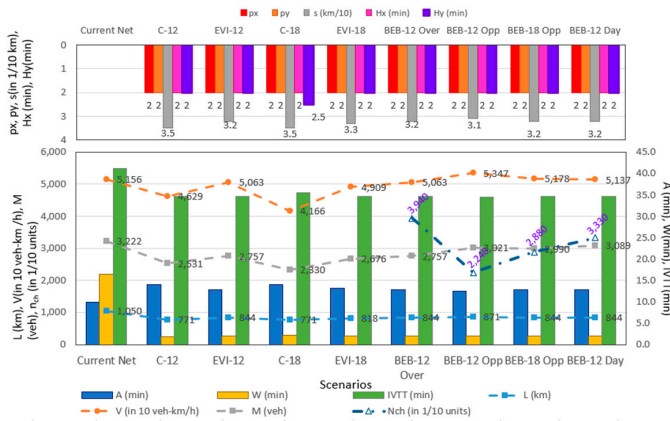

**Figure 4.** Decision variables (**top**), resources and user performance metrics (**bottom**).

### 3.1. Sensitivity Analysis

In this section, it was analyzed how the total cost of the system and the service performance metrics evolve in the different powertrain scenarios according to the steady

changes in the demand, city size and in the unit electric vehicle and unit charging infrastructure costs.

### 3.1.1. Hourly Demand

In Figure 5a, the optimal stop spacing (s), time headway variable (only in horizontal routes, $H_x$) and the total cost of the system (Z) are depicted for different hourly demand values, ranging from 25,000 to 500,000 pax/h for each of the powertrain scenarios.

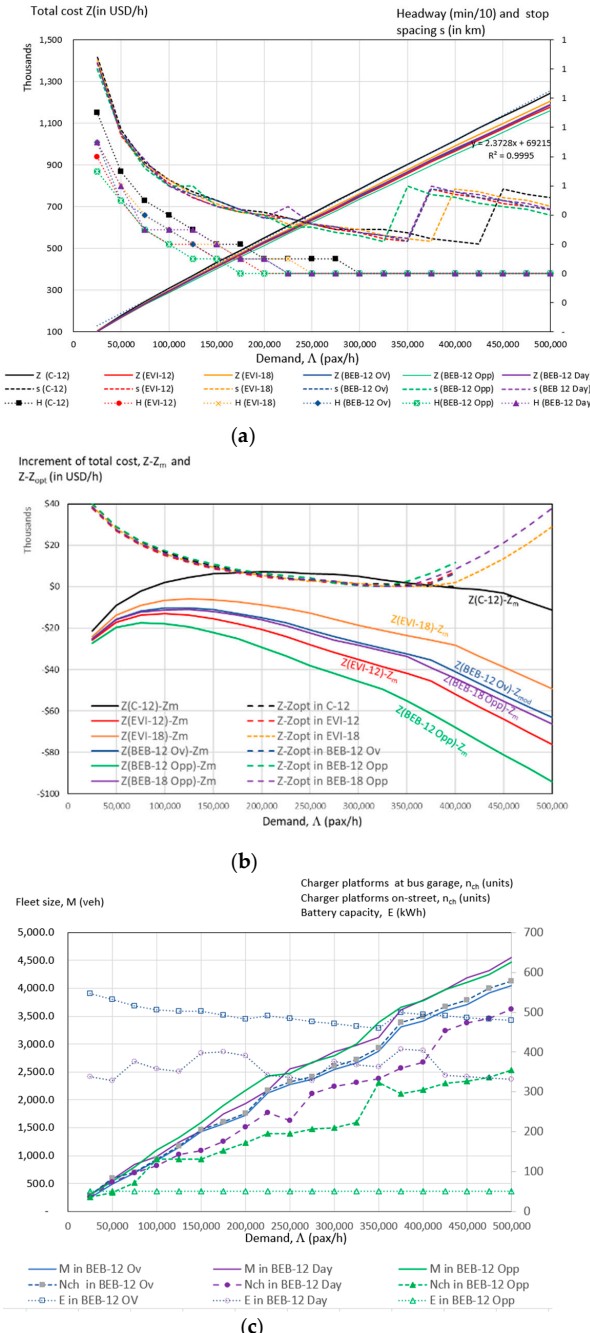

**Figure 5.** (**a**). Total cost and decision variables with regard to the peak hour demand. (**b**). Cost savings of each scenario compared to the Scenario C-12 and the same scenario with spatial variables fixed. (**c**). Fleet size and number of chargers with regard to the peak hour demand.

Figure 5a shows the variation of the optimal network configuration in the total cost (z), the spacing among stops (s) and the headway (H). In general it is noted that all the

powertrain scenarios have a similar performance in terms of a linear increase of the total costs as demand increases. A detail discussion of the results is presented in Section 4.

In order to zoom up in which subdomains each scenario is more efficient, it is depicted in Figure 5b the difference between the total system cost Z in each scenario and the function $Z_m$. The $Z_m$ function was calculated through a linear regression model, considering the total cost of the scenario (C-12) in the corresponding demand data domain of analysis. The resulting analytical expression of this function was $Z_m = 2.3728\Lambda + 69{,}215$.

Finally, Figure 5c shows the results of the variations in the size of the fleet and charging infrastructure required varying the demand.

### 3.1.2. City Size

The same analysis of the bus electric powertrain resources and cost has been repeated when the city size is varied in the domain $D_x \in [3; 30]$ km. The basic assumption was that the city shape and demand density is maintained with regard to the Guadalajara current network, therefore $D_y(km) = 15/18 D_x$ and $\Lambda\left(\frac{pax}{h}\right) = \frac{336000}{15\cdot18} \frac{15}{18} D_x^2$. This sensitive analysis indicates how the size fleet (M), the energy capacity of the battery (E) and the charging numbers changes in functions to the variation of the city size. Figure 6 indicates these results.

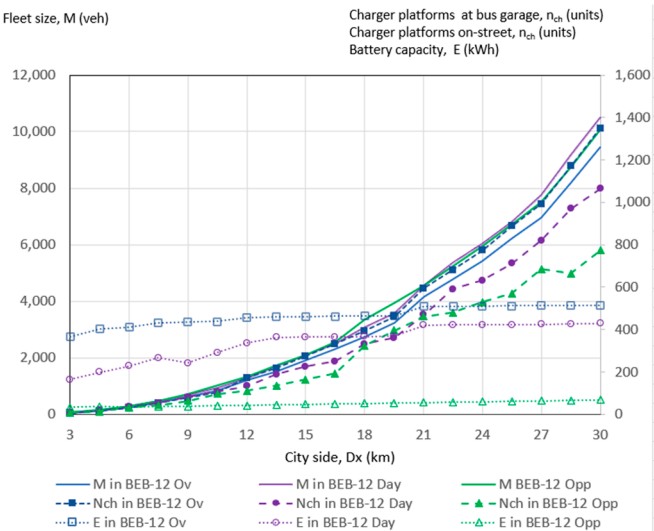

**Figure 6.** Fleet size and number of chargers with regard to the size of the city.

### 3.1.3. Electric Vehicle and Charging Infrastructure Costs

The former efficiency comparison among bus scenarios mainly depends on the cost parameters related to the energy provision, vehicle purchase and the charging/refill facility investment and their depreciation models. Although the acquisition cost of diesel vehicles is quite stabilized, the competition between electric vehicles is evolving rapidly. As a result, it may be expected that the current depreciation cost of electric vehicles $c_M$ is going to decline in the future years. A similar variability may be supposed for the unit vehicle distance cost $c_V$ and the depreciation cost of charging stations $c_N$ for electric powertrains.

Therefore, we will analyze the potential efficiency of electric powertrains in the Guadalajara city for different unit distances and temporal-based vehicle cost domains. Indeed, we will calculate the unit distance cost (USD/veh-km) for electric vehicle of typology x by $(c_V)_{BEB,x} = (c_V)_{EVI,x}(1 + \beta_V)$, x = 12 or 18 m, where $(c_V)_{EVI,x}$ is the corresponding unit distance cost of diesel Euro VI powertrain and $\beta_V$ a reduction factor $(-1 \le \beta_V \le \infty)$. Similarly, it was assumed that the temporal-based cost of electric vehicles (USD/veh-h) can be calculated by $(c_M)_{BEB,x} = (c_M)_{EVI,x}(1 + \beta_M)$, x = 12 or 18 m, where $(c_M)_{EVI,x}$ is the corresponding unit temporal cost of diesel Euro VI powertrain and $\beta_M$ an increment factor $(-1 \le \beta_M \le \infty)$. It should be noted that the new temporal based cost of electric vehicles does not take into account the hourly depreciation cost of the battery pack.

In Figure 7a, it was identified the domain of the factor pair $(\beta_M, \beta_V)$ where the utilization of BEB-12 Ov, Opp and BEB-18 Opp reduces the total cost of the system with regard to the deployment of EVI-12 and EVI-18 respectively. In this case, it was plot the boundary that limits the region where BEB was preferable (bottom-left) for different values of the emission monetization. Hence, we generated the potential monetization values in the future by $\left(\varepsilon_{p,f}\right) = \varepsilon_{p,a}(1 + \beta_\varepsilon)$, where $\varepsilon_{p,a}$ is the current monetization value of pollutant $p$ and $\beta_\varepsilon$ an increment factor $(-1 \leq \beta_\varepsilon \leq \infty)$.

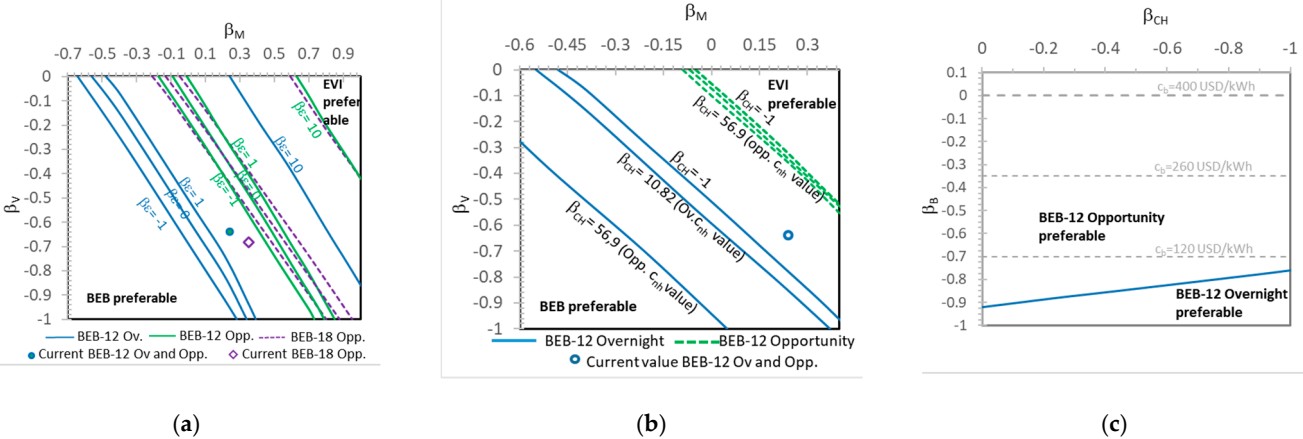

**(a)**     **(b)**     **(c)**

**Figure 7.** Recommended domain for BEB-12 Ov., BEB-12 Opp and BEB-18 Opp technology with regard to (**a**) unit distance, unit temporal cost and emission monetary values, (**b**) unit distance, temporal and charging cost and (**c**) unit charging and battery depreciation cost.

In Figure 7b, a similar analysis was done where the boundary limit among the BEB or EVI effective areas was provided for different depreciation values of the charging station. Finally, in Figure 7c, it was analyzed how much the battery depreciation ($c_B$) and charger cost ($c_n$) should be reduced to economically justify the overnight charging, instead of the opportunity charging scheme.

## 4. Discussions

The results presented in Section 3 can be dived in two main groups of results, which are the performance metrics for all the scenarios proposed, which were presented in Figures 3 and 4 and the sensitive analysis, which were presented in Figures 5–7.

### 4.1. Discussion of the Performance Metrics

The first set of results that summarize the main performance metrics of all the scenarios can be visualized in Figures 3 and 4. Figure 3 contains the total cost of the bus system obtained by the optimization model in different powertrain scenarios and the current system. It can be observed that the main effect is that the modification of the bus network design is the main responsible to lessen the total cost of the system. Only by varying the spatial and temporal configuration to a transfer-based network, the total cost estimated by the model was reduced by 23% with regard to the current network operated by the same vehicles (Scenario C-12). The total system cost barely varied in different bus powertrains under analysis, once the configuration was based on a transfer-based network, instead of a door-to-door network.

The minimal cost of the system was obtained when electric battery standard buses were deployed with an opportunity charging scheme (Scenario BEB-12 Opp). The incremental cost saving of the previous vehicles typology due to the higher powertrain efficiency was 6.02% (net saving between C-12 and BEB-12 Opp). The second-best vehicle is a diesel Euro VI standard bus (Scenario EVI-12), followed by articulated buses charged

in an opportunity operation mode (Scenario BEB-18). The day charging at the bus garage during the day (Scenario BEB-12 Day) was not as competitive as other scenarios.

The charging scheme of standard battery electric buses plays a crucial role in the cost of the system. The opportunity charging scheme of standard BEB vehicles implies more vehicles ($\Delta M$ = 264 veh) and runs more kilometers per hour ($\Delta V$ = 284 veh-km/h) than diesel and overnight-charged electric counterparts. Nevertheless, the cost reduction of BEB-12 Opp with regard to diesel vehicles (EVI-12) was due to the lower unit distance cost parameter ($c_V$). This fact reduced drastically the agency cost component. Other cost components were balanced between diesel and BEB-12 Opp technology. Indeed, the emission cost in Scenario EVI-12 was equivalent to the sum of emission and battery depreciation cost in BEB-12 Opp.

However, the BEB-12 Opp. significantly outperformed the overnight charging scheme due to the lower number of chargers and battery packs. BEB-12 Opp involves $n_{ch}$ = 224 fast charging areas ($N_x$ = 18, $m_x$ = 3, $\phi_x$ = 2, $N_y$ = 29, $m_y$ = 2, $\phi_y$ = 2) and vehicles equipped with a battery pack size of $E$ = 50.96 kWh. On the contrary, the overnight charging scheme will require $n_{ch}$ = 394 areas and a battery pack of $E$ = 463 kWh. Only the depreciation cost of batteries rose to 14% of the total cost, excluding user costs. BEB vehicles with overnight charging scheme would be competitive only if the battery and the garage charging facility cost were reduced. However, due to the battery lifespan (5 years maximum) and the current manufacturing cost, the batteries played a differential role in the competitive analysis. Day charging at the bus garage (BEB-12 Day) requires a similar number of chargers and less battery capacity with regard to overnight charging ($n_{ch}$ = 333 fast ch. areas and $E$ = 367 kWh/veh). Nevertheless, it needs 330 extra vehicles than BEB-12 Ov. due to the death headings to the bus garage and charging time during the service. Moreover, the emission cost becomes a small fraction of the total system expenses in all scenarios. In the current network, C-12 and C-1 scenarios, the monetary value of emissions represents 4–6% of the total cost with the current proxy parameters.

On the other hand, analyzing Figure 4 it can be observed that the optimal time headways and stop spacing vary around $H$ = 2–2.5 min and $s$ = 310–350 m respectively. The existing service in operation outperforms the network accessibility due to the multiple number of routes, which can only be operated at low frequencies (huge waiting times). Hence, the new optimized design reduces waiting times at the expense of rationalizing the number of routes. The optimal bus network configuration is obtained when the stop lattice is $p_x = p_y = 2$, i.e., one out of two consecutive stops allows transfer operation to orthogonal routes. In terms of the amount of total vehicles (M) this shows a variation in the proposed scenarios from the smallest fleet requirement, which was 2330 vehicles in the C-18 scenario to the biggest size requirement, which was 3089 vehicles in the BEB-12 Day scenario. The size variation among the scenario is bonded mainly to the size of the vehicles (some of 12 m and other of 18 m see in Section 2) and the charging scheme.

### 4.2. Discussion of the Sensitivity Analysis

### 4.2.1. Discussion of Sensitivity Analysis of Hourly Demand

Within the hourly demand ($\Lambda$) sensitivity analysis it was observed how total cost of the system (Z), the headway (H) and stop spacing (s) was modified in function to the hourly demand. This result was presented in Figure 5a. The total cost of the system presents an increasing behavior with regard to $\Lambda$, since the user cost component is a linear function of $\Lambda$. While the demand gets more consolidate, the bus route design tends to be operated at lower headways and line spacings. The different behavior in the stop spacing function around $\Lambda$ = 40,000 pax/h is because the optimal stop lattice changes from $p_x = p_y = 2$ ($\Lambda$ < 350,00 pax/h) to be $p_x = 1$ and $p_y = 1$ when $\Lambda$ > 350,00 pax/h. In this domain, the lattice $p_x = p_y = 2$ will need a small stop spacing to accommodate the total demand in the number of corridors, causing a speed reduction. The total cost presents barely the same numerical value in all powertrain scenarios.

In the case of Figure 5b this showed the cost saving for each scenario compared with scenario C-12 when the demand is varying. It can be highlighted from Figure 5b that the powertrain that minimizes the total cost of the system in the whole demand domain is still the BEB-12 Opp vehicle, followed by EVI-12 and electric buses (BEB 18 Opp, 12 m-long overnight and day charging), EVI-18 and C-12.

Moreover, it was also calculated the extra cost that the system will incur when it maintains the spatial bus route configuration during the whole day of service. It was assumed that the bus route configuration was designed for the peak hour in Guadalajara ($\Lambda'$ = 333,613 pax/h) where $p_x = p_y = 2$ and $s$ = 310 m. In the whole domain of potential peak demand $\Lambda \in [25; 500]$ kpax/h, it was optimized the bus network design problem again, maintaining the values of spatial variables $s$, $p_x$ and $p_y$ equal to the former figures and only varying the time headways $H_x$ and $H_y$. This strategy is usually undertaken by transit agencies. The resulting value of the objective function was referred as $Z'$ and it was compared to the value $Z_{opt}$, corresponding to the design problem when all decision variables could be varied. The convex functions $Z'$-$Z_{opt}$ for all scenarios are depicted in Figure 5b by dotted lines. It can be observed that the extra cost of the system was negligible around $\Lambda'$ = 330,000 pax/h (target demand value for the spatial design). It highlights the outstanding robustness of the network design method proposed in this paper. Decision makers may adapt the optimal values of decision variables to the real and available configuration of the street network to draw the final master plan, without incurring a significant extra cost. As the hourly passenger flow from the target demand $\Lambda'$ is increased or decreased, the extra cost increases. When $\Lambda > 400,000$ pax/h, the operation of the prefixed network configuration at the lowest headways in standard buses (12 m long) was not feasible.

Finally, it was analyzed in Figure 5c how the number electric resources (vehicles, battery capacity and chargers) evolved with regard to the hourly demand, maintaining the rest of city and transit agency parameters constant. The battery pack in scenario BEB-12 Ov. varied around $E$ = 500 kWh. Although there are some bus prototypes in the market equipped with batteries of this capacity, this figure exceeds the normal capacity of buses deployed in real service and would become a huge challenge for vehicle manufacturers. On the contrary, the battery pack needed in scenario BEB-12 where vehicles are charged at terminal stops presents a constant value of $E$ = 51.0 kWh. The battery pack in the BEB-12 Day scenario, remained stable around 350 kWh.

The number of charging stations increased as the demand consolidates and as a result, more vehicles were needed. Nevertheless, the ratio $M/n_{ch}$ slightly increased as the demand growths. It remained stable in each charging scheme around 6–12 buses per charger for the whole domain of service.

### 4.2.2. Discussion of Sensitivity Analysis of City Size

In the case of the variation of the city size this approach was made to analysis the variation in the fleet size (M), the charging infrastructure required ($N_{ch}$) and the battery capacity (E) for each electric scenario. This result was presented in Figure 6.

The battery pack in scenario BEB-12 Opp presents in Figure 6 a linear increment with regard to $D_x$, ranging from $E_{bat}$ = 34 to 68 kWh in the domain of analysis. This was an expected tendency since the consumption between two consecutive chargers in horizontal and vertical corridors are respectively $D_x \cdot f_c$ and $\frac{15}{18} D_x \cdot f_c$. However, in all scenarios, the number of charging stations, the number of vehicles and consequently, the depreciation cost of batteries and charging facilities, do not present the linear growth with regard to $D_x$ variable as happened in Figure 5c. This fact is the result of the hourly demand that grows quadratically with $D_x$.

### 4.2.3. Discussion of Sensitivity Analysis of Electric Vehicle and Charging Infrastructure Costs

Finally, in the last set of results of Section 3 a sensitivity analysis was made varying cost in charging infrastructure and vehicles. These results were presented in Figure 7a–c.

In Figure 7a, when the $c_V$ parameter (units distance cost) in BEB and Euro-VI vehicles was the same ($\beta_V = 0$) and current emission monetization ($\beta_\varepsilon = 0$), BEB-12 Ov. and BEB-12 Opp. vehicles were only cost competitive if its vehicle depreciation cost was reduced by 60% and 15% respectively, with regard to the corresponding EVI-12 cost. If the emission monetization values are doubled ($\beta_\varepsilon = 1$), the domain of ($\beta_{MB}, \beta_V$) where BEB is cost-competitive grows. Nevertheless, the domain increment was marginal. If externalities are monetized 10 times higher, the BEB technology is cost efficient even when the depreciation cost of electric vehicles is higher than diesel counterparts. On the contrary, the situation when emissions were not considered in the objective function ($\beta_\varepsilon = -1$) was analyzed, the domain of successful ($\beta_{MB}, \beta_V$) for the deployment of electric vehicles was reduced.

In the case of the 7b case, the charging station cost for BEB-$x$ ($x = 12$ m and 18 m) was given by $(c_n)_{BEB,x} = (c_n)_{EVI}(1 + \beta_{ch})$, where $(c_n)_{EVI} = 0.104$ USD/charger-h was the depreciation cost of the fuel dispenser station per hour and diesel vehicle and $\beta_{ch}$ the increment factor. As it can be noticed, any increment of $\beta_{ch}$ reduced the effective area where the BEB outperformed the diesel Euro-VI counterpart. Nevertheless, the ($\beta_M, \beta_V$) boundary was practically insensitive to any change in $\beta_{ch}$ in BEB-12 Opp.

Finally the results showed in Figure 7c and given the current parameters defined in Table A1 for Guadalajara city, the battery cost reduction was calculated by $(c'_B)_{new} = (c_B)_R(1 + \beta_B)$ for both overnight and opportunity charging schemes, where $(c_V)_R = 0.019$ USD/kWh-h is the current battery depreciation cost assuming a purchasing cost of USD 400/kWh and $\beta_B$ the corresponding reduction factor ($-1 \leq \beta_B \leq \infty$). The reduction in the charging infrastructure cost was modeled by the reduction factor $\beta_{ch}$ defined before. It can be noticed from Figure 7c that the recommended domain for BEB-12 Overnight technology was really sensitive to the depreciation cost of batteries. This cost must be reduced by more than 75% ($\beta_B > 0.75$) to justify the implementation of overnight charged vehicles with battery packs higher than $E = 450$ kWh. It means that the purchasing cost of batteries should be even lower than the most favorable scenario of 120 USD/kWh [45].

## 5. Conclusions

This paper presents a robust methodology to design efficient bus networks operated by battery electric buses or combustion engine buses. This aspect of the paper is the main contribution of the authors to the knowledge of the field. Only defining a set of parameters addressing the shape of the city, demand attributes and capital-operating costs of the vehicle technology, the proposed model directly calculates the optimal bus service design and the costs incurred by agency, users and environment in terms of emissions. Although the model applicability and charging performance was tested in Guadalajara (México), the methodology is easily transferable to any city with a grid-shaped street network.

In the Guadalajara test instance, the optimal design based on a grid network is responsible for the reduction of the current total cost of the existing door-to-door network by 27%, properly balancing the spatial and temporal variables. It is important to highlight that this reduction corresponds to an analytical model based on continuous approximations. The outstanding total cost savings obtained are similar to the corresponding other cities where the bus network design was switched from direct to transfer-based trips [28]. The magnitude of the cost savings, resulted from a new network design, is higher than the ones that can be obtained when the vehicle technology is replaced by more efficient powertrains. This is due to the lower agency cost (less infrastructure length and reduced fleet size) required by transfer-based schemes, compared to the current system. Moreover, the low pollutant proxies caused the external cost contribution in the system cost to be marginal. Indeed, the external cost represented less than 7% of the total cost in fossil fuel powertrains, while in electric powertrains this figure was less than 1%. This potential saving in the externality cost component in favor of BEB was neutralized by the current expensive purchasing price of electric vehicles and batteries. As a result, the total cost of the system did not vary significantly among the considered scenarios in the optimized network,

reproducing different bus powertrains. In addition to that, the network design (spatial and temporal coverage) was quite stable in all powertrains.

However, the results obtained by 12 m-long BEB with the opportunity charging scheme outperformed the others, although it usually implies more vehicles due to the charging time spent on-route. The second-best powertrains were EVI-12, followed by BEB-18 and BEB-12 Ov and BEB-12 Day. This ranking is maintained for a wide domain of demand, considering the current purchasing and maintenance cost of resources. However, considering the sensitivity analysis performed, the increment of the emission monetization would allow a greater justification of BEB technology, even if the vehicle temporal cost parameter was currently higher than the alternative EVI technology counterpart. The variation of the facility cost of charging stations practically did not have a high impact on the domains of other cost parameters. Another thing to be aware is that the monetization costs for local pollutants are given as average values for countries in terms of Euros/vehicle-kilometer, and do not consider the population density in the areas where transport projects are going to be deployed. Therefore, the temporal cost component of the objective function is affected by the high demand (as in the case of Guadalajara), while the negative health effects on the population only depends on the fleet mileage. The authors recommend the estimation of more realistic and case-sensitive external cost parameters to be balanced with other cost components.

From the vehicle design perspective, the performance of BEB-12 overnight technology is questionable due to the thermal management, weight and space needed on-board by the huge battery pack. The cost of this extremely large battery pack reduces the efficiency of this technology in favor of the opportunity charging alternative. The day charging at bus garage in service is only cost-efficient if the unproductive dead-heading travel time to bus garages are negligible. Indeed, in the BEB-12 Opp. scenario, the battery pack presents a treatable size that minimizes the time spent at chargers.

Nevertheless, the major challenge of the opportunity charging scheme is the construction of on-street charging stations. The best solution for these transit services has been obtained when routes are operated with low time headways. In order to completely charge the battery pack at the terminal station and allow some headway disruptions, each charging station will need two or more bus areas (platforms) to charge simultaneously multiple vehicles at the same time. This proposal is questionable from the city perspective, due to the public space consumption. Due to the space scarcity in the city areas, several authorities are reluctant to allocate areas in the sidewalks to install charging facilities. Moreover, the same authorities also argue that the on-street charging stations reduce the flexibility of the bus system, since routes cannot be extended or modified. In spite of the outstanding cost-efficiency of the opportunity charging scheme, there are still several controversial issues to be solved to clearly promote this charging operation.

Therefore, the adoption of BEB technology presents several challenges due to the higher depreciation cost of batteries and vehicles, and additional number of charging resources. Decision makers and transit agencies must evaluate the new incremental resources and even mileage caused by electrification in order to maintain the target service in terms of headways and the objectives of urban mobility plans.

**Author Contributions:** All authors contributed equally to this work. All authors have read and agreed to the published version of the manuscript.

**Funding:** Funding to the elaboration of this work was done with the support of the Mexican National Council for Science and Technology (CONACYT) and the Universitat Politècnica de Catalunya (UPC).

**Institutional Review Board Statement:** Not applicable.

**Informed Consent Statement:** Not applicable.

**Data Availability Statement:** Not applicable.

**Acknowledgments:** The authors are grateful with because the technical and financial support made by the Universitat Politècnica de Catalunya (UPC) and the Mexican National Council for Science and Technology (CONACYT).

**Conflicts of Interest:** The authors declare no conflict of interest.

## Appendix A

**Length of the transport network,** $L$: The number of vertical lines to be implemented in the city can be estimated by means of the quotient $\frac{D_x}{s_x}$, knowing that each route has a length equal to $D_y$. Similarly, the number of horizontal lines is $\frac{D_y}{s_y}$, each having a length of $D_x$. The network length can be easily calculated as the number of vertical and horizontal lines multiplied by their corresponding lengths, according to Equation (A1).

$$L = L_x + L_y = \frac{D_y D_x}{s_y} + \frac{D_x D_y}{s_x} = D_y D_x \left( \frac{1}{p_y s} + \frac{1}{p_{xy} s} \right) \tag{A1}$$

**Distance travelled by the fleet in one hour of service,** $V$: In each horizontal line, there is a vehicle that will run a complete round trip in every time headway $H_x$. If all horizontal lines are considered, the total distance travelled by the fleet in the Cartesian component $x$ will be equal to twice (double direction) the total length of the horizontal corridors $L_x$ divided by their step interval $H_x$ (Equation (A2a)). Nevertheless, in the opportunity charging scheme, we have to compute the deadheading distance of all vehicles operating horizontal corridors that must stop at the electric charging station, $L_{ch,x}$. The resulting formula is given in Equation (A2b). The horizontal component of deadheading distance in one corridor along x-axis is given by $l_{ox}$. The vertical component $d_v$ depends on the relative spacing between horizontal corridors and the spatial distribution of charging stations in the E-W periphery sectors of the city (see Figure 1). If charging is evenly distributed and centered along the city side $D_x$, this distance can be approximated by $d_v \cong \frac{D_y}{4 N_x}$ when $N_y < \frac{D_y}{s_y}$. Nevertheless, this vertical distance is considered to be 0 in all horizontal corridors when there is a charger station at the terminal stop of each route ($d_v = 0$ if $N_y = \frac{D_y}{s_y}$). The aggregation of this component for all horizontal lines can be calculated by $\sum_{i=1}^{D_y/s_y} d_v \cong \frac{D_y}{4 N_x} \frac{D_y}{s_y}$ Therefore, the estimation of the total deadheading length is $L_{ch,x} = \frac{D_y}{s_y} l_{ox} + d_v \frac{D_y}{s_y}$. An identical approach can be developed for vertical corridors, obtaining $L_{ch,y} = \frac{D_x}{s_x} l_{oy} + d_h \frac{D_x}{s_x}$. In the former expression, $d_h \cong \frac{D_x}{4 N_y}$ when $N_y < \frac{D_x}{s_x}$. In the case of day charging at the bus garage during the service, each vehicle of the fleet operating horizontal corridors will travel a distance equivalent to $2 l_G$ every charging cycle, i.e., every $\sum_{i=0}^{\pi_x} \Delta t_{i,x}$ units of time. The fleet size required to operate the horizontal routes without charging operations is given by $\frac{2 L_x}{v_{c,x} \cdot H_x}$, as it is proved in Equation (A5). Therefore, the extra distance run by the horizontal fleet in one hour due to the dead heading movements to the bus garage is given by $\frac{2 L_x}{v_{c,x} \cdot H_x} \frac{2 l_G}{\sum_{i=0}^{\pi_x} \Delta t_{i,x}}$ Similar expression can be found for vertical corridors. Finally, the estimation of the total distance run by the whole fleet in one hour in the BEB-12 day scenario is made by Equation (A2c).

$$V = V_x + V_y = \frac{2(L_x)}{H_x} + \frac{2(L_y)}{H_y} = 2 D_y D_x \left( \frac{1}{H_x p_y s} + \frac{1}{H_y p_x s} \right) \tag{A2a}$$

$$\begin{aligned} V = \frac{2(L_x + L_{ch,x})}{H_x} &+ \frac{2(L_y + L_{ch,y})}{H_y} \\ &= 2 D_y D_x \left( \frac{1}{H_x p_y s} + \frac{1}{H_y p_x s} \right) + \frac{2 D_y}{H_x p_y s}(l_{ox} + d_v) + \frac{2 D_x}{H_y p_x s}(l_{oy} + d_h) \end{aligned} \tag{A2b}$$

$$V = \frac{2(L_x)}{H_x} + \frac{2(L_x)}{v_{c,x}H_x}\frac{2l_G}{\sum_{i=0}^{\pi_x}\Delta t_{i,x}} + \frac{2(L_y)}{H_y} + \frac{2(L_y)}{v_{c,y}H_y}\frac{2l_G}{\sum_{i=0}^{\pi_y}\Delta t_{i,y}}$$

$$= 2D_yD_x\left(\frac{1}{H_xp_ys} + \frac{1}{H_yp_xs}\right) + \frac{2D_yD_x}{v_{c,x}H_xp_ys}\left(\frac{2l_G}{\sum_{i=0}^{\pi_x}\Delta t_{i,x}}\right) \quad \text{(A2c)}$$

$$+ \frac{2D_yD_x}{v_{c,y}H_yp_xs}\left(\frac{2l_G}{\sum_{i=0}^{\pi_y}\Delta t_{i,y}}\right)$$

**Average number of transfers**, $p(1)$: Let us assume that there is a user's trip that originated at point O1 of Figure A1a. The user should not make transfers if its destination is located within a longitudinal band of width $p_ys$ or a vertical band of width $p_xs$, both centered at point O1. The corresponding area of these bands is $D_xp_ys + D_yp_xs$. Thus, the probability of making zero transfers is equivalent to the probability that the travel destination D1 is contained within these bands. Assuming that origins and destinations of trips are uniformly distributed in the network, the probability of making 0 transfers is equivalent to the ratio between the area of the aforementioned bands and the total area of the city. However, the area resulted from the intersection of the two bands will be subtracted from the numerator to avoid double-counting. Finally, the probability of making a transfer is easily determined knowing that $p(0) + p(1) = 1$. In this way, the expected number of transfers is equivalent to the probability of making 1 transfer, $p(1)$, defined in Equation (A3).

$$p(1) = 1 - \frac{p_xsD_y + p_ysD_x - p_xp_ys^2}{D_xD_y} \quad \text{(A3)}$$

**Net commercial speed**, $v_c$: The pace of vehicles, i.e., the inverse of the net commercial speed, is calculated as the summation of the time needed to overcome one kilometer at the maximal cruising speed $v$, the additional time for braking and accelerating the vehicle before/after all stops located in one kilometer, and the boarding and alighting time of users in one kilometer. Note that in the latter component, the number of boarding passengers along horizontal corridors also embraces those passengers that transferred from vertical to this horizontal route $p(1)\Lambda/2$. This figure is divided by the distance run in service per hour, $V_x$ (the one given by Equation (A2a)) to calculate the boarding passengers on one vehicle along one kilometer. These three components correspond to the first three terms of Equation (A4) and are also calculated in Estrada [25]. Nevertheless, the fourth term consists of the layover time at the two terminal stops of the horizontal ($\theta_W, \theta_E$) and vertical routes ($\theta_N, \theta_S$). Moreover, the fifth term is the positive difference between the time spent at charging vehicles at the terminal stops and the previous lay-over time in the case of opportunity charging. If the charging time is higher than the layover time, vehicles are held for a longer time at terminal stops, becoming less productive and shrinking the net commercial speeds.

$$\frac{1}{v_{c,x}} = \frac{1}{v} + \frac{1}{s}\tau + \frac{\Lambda}{2V_x}[1 + p(1)]\tau' + \frac{\theta_E + \theta_W}{2(L_x + L_{ch,x})} + \frac{\max\{0, T_{c,E} - \theta_E\} + \max\{0, T_{c,W} - \theta_W\}}{2(L_x + L_{ch,x})} \quad \text{(A4a)}$$

$$\frac{1}{v_{c,y}} = \frac{1}{v} + \frac{1}{s}\tau + \frac{\Lambda}{2V_y}[1 + p(1)]\tau' + \frac{\theta_S + \theta_N}{2\left(L_y + L_{ch,y}\right)} + \frac{\max\{0, T_{c,S} - \theta_S\} + \max\{0, T_{c,N} - \theta_N\}}{2\left(L_y + L_{ch,y}\right)} \quad \text{(A4b)}$$

**Number of vehicles required to operate the service in one hour**, $M$: The size of the fleet can be easily calculated by means of Equation (A5), breaking it down into its horizontal ($M_x$) and vertical component ($M_y$). The number of kilometers operating the horizontal corridors can be directly calculated by the quotient between the number of kilometers run along horizontal corridors in one hour ($V_x$) and the net commercial speed ($v_{c,x}$). A similar approach can be done to calculate the fleet size in vertical corridors. In the case of opportunity charging we have to consider the time spent travelling the unproductive distance to access the chargers $L_{ch,x}$ and $L_{ch,y}$. In the case of the day charging at the bus garage, we consider the extra fleet needed in each horizontal and vertical route ($\Delta\mu_x$ and $\Delta\mu_y$) multiplied by the number of horizontal and vertical corridors respectively.

$$M = M_x + M_y = \frac{2\left(L_x+L_{ch,x}\right)}{v_{c,x}H_x} + \Delta\mu_x\frac{D_y}{p_y s} + \frac{2\left(L_y+L_{ch,y}\right)}{v_{c,y}H_y} + \Delta\mu_y\frac{D_x}{p_x s} =$$
$$= \frac{2D_y}{v_{c,x}H_x p_y s}\left(D_x + l_{ox} + d_v\right) + \frac{2D_x}{v_{c,y}H_y p_x s}\left(D_y + l_{oy} + d_h\right) + \Delta\mu_x\frac{D_y}{p_y s} + \Delta\mu_y\frac{D_x}{p_x s} \tag{A5}$$

**Access time**, *A*: This component will depend on the spacing of the stops and the transport lines $(p_x, p_y)$. We assume that the probability to board on a vertical and horizontal corridor is equal. Therefore, access distance is calculated (see Figure A1b) as one-half of the summation of the average distance to a horizontal corridor $a_H$ (from brown coloured station) and to a vertical corridor $a_V$ (from green colored stations). This distance is considered twice to reflect the access and egress leg in the whole trip and it is divided by the average walking speed to finally estimate the access time through Equation (A6).

$$A = 2\frac{a_H + a_V}{2v_w} = s\left(\frac{2 + p_x + p_y}{2v_w}\right) \tag{A6}$$

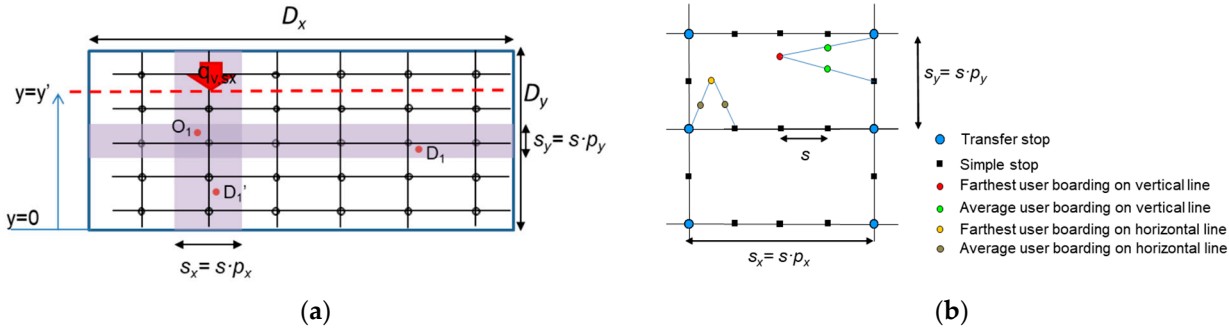

(**a**)　　　　　　　　　(**b**)

**Figure A1.** (**a**). Scheme of the spatial coverage area of routes (shadowed) and passenger vertical flow. (**b**). Scheme of the stop lattice and access distances.

**Average waiting time at stop**, *W*: This variable considers the average total time spent by a user on the stops, waiting for the arrival of the transport vehicle. This network allows covering all the trips of the city by means of a transfer, at most. Therefore, the waiting time is computed by $W = p(0)\cdot w(0) + p(1)\cdot w(1)$, where $w(x)$ and $p(x)$ are the waiting time and the probability to make $x$ transfers respectively. Assuming an average headway $\widetilde{H}$ in routes, the waiting time for users who do not make transfers is $w(0) = \widetilde{H}/2$, while the waiting time for trips with one transfer includes the waiting time in the first stop $(\widetilde{H}/2)$ and the corresponding at the transfer stop $(\widetilde{H}/2)$. In this last case, we assume that the arrivals of the vehicles of horizontal and vertical lines are not coordinated in the transfer stops. In this way, the average waiting time is defined by Equation (A7), considering $\widetilde{H} = \frac{H_x+H_y}{2}$.

$$W = \frac{p_x s D_y + p_y s D_x - p_x p_y s^2}{D_x D_y} \times \frac{H_x+H_y}{4}$$
$$+ \left(1 - \frac{p_x s D_y + p_y s D_x - p_x p_y s^2}{D_x D_y}\right)\frac{H_x+H_y}{2} \tag{A7}$$

**Transfer time**, $T_{TR}$: By transfer time we will understand the time needed to travel on foot from the loading area where the user gets off the first vehicle to the new loading area where the departing bus will stop, to continue his/her journey. The waiting time at the transfer stop is already considered in variable *W*. The transfer stops will be considered to have a proper design so that the user must walk a distance $\delta$ at speed $v_w$ In this way, the transfer time can be easily calculated by Equation (A8), considering that only a fraction $p(1)$ of all users will make a transfer to complete their trip.

$$T_{TR} = \frac{\delta \cdot p(1)}{v_w} = \frac{\delta}{v_w}\left(1 - \frac{p_x s D_y + p_y s D_x - p_x p_y s^2}{D_x D_y}\right) \tag{A8}$$

**In-vehicle travel time**, *IVTT*: As we assume the L1 distance metric, the expected distance between the origin and the destination point (O1 and D1 respectively) of an average trip will be calculated independently in the two Cartesian components: $(d_{IVVT})_x$ and $(d_{IVVT})_y$. For the sake of simplicity, we will only focus on the *x*-component $(d_{IVVT})_x$. First of all, we assume that the location of point O1 is fixed at x = $x_O$ and its distance to the destination D1 is denoted by $(d_{IVVT})_{x,O}$. The distance between points O1 and D1, given that the point O is located at $x = x_o$ is $(d_{IVTT})_{x,o} = p(x_D < x_O)d_{xD<xO} + p(x_D \geq x_O)d_{xD\geq xO}$ where $p(x_D < x_O)$ is the probability that point D1 falls on the left side of the origin O1 and $d_{xD<xO}$ the distance between these two points when $x_D < x_O$. As origins and destinations of users are uniformly distributed throughout the city, the probability $p(x_D < x_O)$ can be calculated as the ratio between the effective length of the segment in which D1 can be located on the left side of O1 $(x_O/D_x)$, while the distance $d_{xD<xO}$ is calculated as one half of the effective length of the segment in which D1 can be located $(x_O/2)$. The probability corresponding to the situation that D1 is located on the right side of O can be estimated in a similar way, obtaining the following results: $p(x_D \geq x_O) = (D_x - x_O)/D_x$ and $d_{xD\geq xO} = (D_x - x_O)/2$. With these results we can obtain $(d_{IVTT})_{x,o} = \frac{x_{O1}^2}{2D_x} + \frac{(D_x - x_{O1})^2}{2D_x}$.

However, the origin point O1 can be fixed at any point on the *x*-axis and the destination point can be located on both, its right and left sides. The expected distance in the component *x* of the evenly distributed points O1 and D1 can be estimated by $(d_{IVTT})_x = \frac{1}{D_x}\int_0^{D_x}(d_{IVTT})_{x,o}dx_O = D_x/3$. Similarly, the distance for the vertical component is estimated by $(d_{IVTT})_y = D_y/3$. Finally, the travel time inside the vehicle can be estimated by means of Equation (A9), where $v_c'$ is the gross commercial speed of public transport vehicles. The gross commercial speed is equal for all bus powertrains and can be calculated by Equation (11a,b) letting $\theta_S = \theta_N = \theta_E = \theta_W = T_{c,S} = T_{c,N} = T_{c,E} = T_{c,W} = l_{ox} = d_v = l_{oy} = d_h = 0$.

$$IVTT = \frac{D_x}{3(v_c')_x} + \frac{D_y}{3(v_c')_y} \tag{A9}$$

**Occupancy**, *O*: We aimed at calculating the section $y = y'$ that presents a maximum vehicle occupancy in a vertical route like the one depicted in Figure A1. It is assumed that one half of the users are going to complete their trip by accessing a vertical line (although later they can make a transfer to a horizontal line), and the other half will do in a downstream direction. Therefore, the total flow of passengers downwards throughout the width of the city is $Q_V = \Lambda(1 + p(1))/4$. The total downward total vertical flow can be calculated by the former flow $Q_V$ divided by the total number of lines, i.e., $q_{V,sx} = Q_V \cdot s_x/D_x$. Finally, it is necessary to determine the passenger flow crossing a generic section $y = y'$ (discontinuous red line in Figure A1). For a user crossing the section $y = y'$, the coordinate of its origin $(y_O)$ must satisfy $y_O > y'$, while the coordinate of its destination $(y_D)$ must verify $y_D < y'$. Since the origins and destinations of users are evenly distributed along $D_y$, the probability that the origin is located in sections $y_O > y'$ is equivalent to $p(y_O > y') = \frac{D_y - y'}{D_y}$. Similarly, the probability that its destination fulfills $y_D < y'$ is $p(y_D < y') = \frac{y'}{D_y}$. Finally, the probability that a vertical trip downwards will cross the section $y = y'$ is given by $p(y_O > y'|y_D < y') = \frac{D_y - y'}{D_y}\frac{y'}{D_y}$. Therefore, the passenger flow at one route at section $y = y'$ is defined by $q_{V,s_x}(y') = Q_V s_x \frac{D_y - y'}{D_x D_y}\frac{y'}{D_y}$. It can be demonstrated that the maximal flow is obtained when $y' = D_y/2$, obtaining $(y_O > D_y/2|y_D < D_y/2) = \frac{1}{4}$. In this way, the value of the occupancy at section $y' = D_y/2$ in a vertical line $O_y$ can be estimated by Equation (A10), as the product of the flow $q_{V,s_x}(y')$ and the time headway $H_y$. This occupancy must

be less than or equal to the vehicle capacity. The occupancy in horizontal lines can be calculated in a similar way.

$$O_y = \frac{\Lambda}{16}(1 + p(1))\frac{s_x}{D_x}H_y \tag{A10}$$

## Appendix B

**Table A1.** Monetary values and emission factors corresponding to each pollutant.

| | | $CO_2$ | $PM_{10}$ | $NO_X$ | CO | $SO_X$ | VOC | $NH_3$ |
|---|---|---|---|---|---|---|---|---|
| C-12 | Monetization$_x$ (USD/g polluntant $x$) | $1.12 \times 10^{-4}$ | $1.38 \times 10^{-1}$ | $2.39 \times 10^{-2}$ | $1.34 \times 10^{-3}$ | $1.22 \times 10^{-2}$ | $1.34 \times 10^{-3}$ | $1.96 \times 10^{-2}$ |
| | Tank to Wheel, $E_{F,x}$ (g/km) | $1.67 \times 10^{3}$ | $1.21$ | $2.23 \times 10^{1}$ | $8.42$ | $3.14 \times 10^{-3}$ | $2.88$ | $2.90 \times 10^{-3}$ |
| | Well to Tank, $E_{E,x}$ (g/kWh) | $5.51 \times 10^{1}$ | $1.13 \times 10^{-2}$ | $9.76 \times 10^{-2}$ | $4.49 \times 10^{-2}$ | $3.27 \times 10^{-2}$ | $2.63 \times 10^{-2}$ | |
| | Vehicle manufacturing, $E_{M,x}$ (g/veh-h) | $1.43 \times 10^{3}$ | | | | | | |
| | Infrastructure, $E_{I,x}$ (g/km-h) | $6.19 \times 10^{2}$ | | | | $3.24$ | | |
| Euro-VI 12 | Tank to Wheel, $E_{F,x}$ (g/km) | $1.28 \times 10^{3}$ | $8.11 \times 10^{-3}$ | $5.28 \times 10^{-1}$ | $3.64 \times 10^{-1}$ | $2.40 \times 10^{-3}$ | $5.70 \times 10^{-2}$ | $9.00 \times 10^{-3}$ |
| | Well to Tank (g/kWh) | $5.51 \times 10^{1}$ | $1.13 \times 10^{-2}$ | $9.76 \times 10^{-2}$ | $4.49 \times 10^{-2}$ | $3.27 \times 10^{-2}$ | $2.63 \times 10^{-2}$ | |
| | Vehicle manufacturing, $E_{M,x}$ (g/veh-h) | $1.43 \times 10^{3}$ | | | | | | |
| | Infrastructure, $E_{I,x}$ (g/km-h) | $6.19 \times 10^{2}$ | | | | $3.24$ | | |
| C-18 | Tank to Wheel, $E_{F,x}$ (g/km) | $2.07 \times 10^{3}$ | $1.52$ | $2.87 \times 10^{1}$ | $1.11 \times 10^{1}$ | $3.89 \times 10^{-3}$ | $2.99$ | $2.90 \times 10^{-3}$ |
| | Well to Tank, $E_{E,x}$ (g/kWh) | $5.51 \times 10^{1}$ | $1.13 \times 10^{-2}$ | $9.76 \times 10^{-2}$ | $4.49 \times 10^{-2}$ | $3.27 \times 10^{-2}$ | $2.63 \times 10^{-2}$ | |
| | Vehicle manufacturing, $E_{M,x}$ (g/veh-h) | $1.93 \times 10^{3}$ | | | | | | |
| | Infrastructure, $E_{I,x}$ (g/km-h) | $6.19 \times 10^{2}$ | | | | $3.24$ | | |
| Euro VI 18 | Tank to Wheel, $E_{F,x}$ (g/km) | $1.70 \times 10^{3}$ | $9.15 \times 10^{-3}$ | $4.38 \times 10^{-1}$ | $4.14 \times 10^{-1}$ | $3.19 \times 10^{-3}$ | $6.5 \times 10^{-2}$ | $9.0 \times 10^{-3}$ |
| | Well to Tank, $E_{E,x}$ (g/kWh) | $5.51 \times 10^{1}$ | $1.13 \times 10^{-2}$ | $9.76 \times 10^{-2}$ | $4.49 \times 10^{-2}$ | $3.27 \times 10^{-2}$ | $2.63 \times 10^{-2}$ | |
| | Vehicle manufacturing, $E_{M,x}$ (g/veh-h) | $1.93 \times 10^{3}$ | | | | | | |
| | Infrastructure, $E_{I,x}$ (g/km-h) | $6.19 \times 10^{2}$ | | | | $3.24$ | | |
| BEB 12 Ov and Opp. | Tank to Wheel, $E_{F,x}$ (g/km) | | | | | | | |
| | Well to Tank, $E_{E,x}$ (g/kWh) | $5.14 \times 10^{2}$ | $8.58 \times 10^{-2}$ | $8.02 \times 10^{-1}$ | $2.55 \times 10^{-1}$ | $6.90 \times 10^{-1}$ | $6.29 \times 10^{-2}$ | |
| | Vehicle manufacturing, $E_{M,x}$ (g/veh-h) | $1.79 \times 10^{3}$ | | | | | | |
| | Infrastructure, $E_{I,x}$ (g/km-h) | $6.19 \times 10^{2}$ | | | | $3.24$ | | |
| | Chargers (g/charger), $E_{C,x}$ (g/charger-h) | $1.12 \times 10^{2}$ | $1.04 \times 10^{-6}$ | $1.24 \times 10^{-1}$ | $1 \times 10^{-1}$ | $1.16 \times 10^{-1}$ | $8.16 \times 10^{-6}$ | $2.78 \times 10^{-6}$ |
| BEB-18 Opp | Tank to Wheel, $E_{F,x}$ (g/km) | | | | | | | |
| | Well to Tank, $E_{E,x}$ (g/kWh) | $5.14 \times 10^{2}$ | $8.58 \times 10^{-2}$ | $8.02 \times 10^{-1}$ | $2.55 \times 10^{-1}$ | $6.90 \times 10^{-1}$ | $6.29 \times 10^{-2}$ | |
| | Vehicle manufacturing, $E_{M,x}$ (g/veh-h) | $2.42 \times 10^{3}$ | | | | | | |
| | Infrastructure, $E_{I,x}$ (g/km-h) | $6.19 \times 10^{2}$ | | | | $3.24$ | | |
| | Chargers (g/charger), $E_{C,x}$ (g/charger-h) | $1.12 \times 10^{2}$ | $1.04 \times 10^{-6}$ | $1.24 \times 10^{-1}$ | $1 \times 10^{-1}$ | $1.16 \times 10^{-1}$ | $8.16 \times 10^{-6}$ | $2.78 \times 10^{-6}$ |

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
