# Peer review of "Battery Electric Bus Network: Efficient Design and Cost Comparison of Different Powertrains"

_sustainability, doi:10.3390/su13094745_

Round 1

Reviewer 1 Report

Dear Authors,

Thank You for the opportunity of reading this article. My general opinion about the presented range of the article is positive.

General statements about the article:

-> article proposes s a methodology to design an efficient transit network operated by battery electric buses in cities with grid-shaped road  network, based on continuous approximations.  So the topic of research is actual and desirable.

-> The article content suite to Sustainability journal scope

-> the abstract is adequate to article content

-> Keywords are correctly proposed.

-> Methods are clearly introduced.

-> The results are based on case study of Guadalajara (Mexico), that increase the value of the results.

-> Literature review is based on 50. They are related to article content.

I indicate the one issue that requires additional clarification:

#1

Generally, the literature state of the art is clearly realized in section 1. It’s well written and adequate. But no positions from 2020 or 2021 are not included. So please extend with at least 3 positions from 2020 or 2021 to update the value of the state of the art

#2

The results in the article are well presented. This is a strong element of the article but there is a lack of discussion. Thus please add a separate section with discussion.

# 3

Please also revise the manuscript regarding the personal way of addressing in the text. Please avoid and replace we" or "our" with the impersonal manner of addressing. The text will sound much more professional.

Technical issues:

-> Please improve the quality of figs 5, 6, 7.

-> Additionally please add some elements that are required by journal like author contribution or data availability statements etc.

Author Response

Dear reviewer 1,

Thank you very much for your review. We have made the changes according to your comments in the revised version. Please kindly check.

Reviewer 2 Report

I have the impression that the paper has already been subjected to careful revision.
The subdivision is handled very well.
The topics are well chosen.
The method is well described.
The case study is significant.
The conclusions are very extensive.
there is an appendix that completes everything.

Author Response

Dear reviewer 2,

Thank you very much for your review. We have made the changes according to your comments in the revised version. Please kindly check.

Reviewer 3 Report

The authors present a very complex and detailed model to compare the choice of different power trains in the operation of a large bus fleet in a Mexican city. While some of the aspects they address are very detailed, I think there are some major shortcomings that need to be addressed.

I have two major concerns on some crucial hypotheses at the basis of the model that is presented:

  1. Line 145 states that “origins 145 and destinations uniformly distributed in the service area Dx and Dy”: this seems a very strange hypothesis, since research works are usually based on OD matrices (as mentioned by the authors in their literature review) that are far from representing a uniform distribution of origins and destinations. It is clear that such an approximation is a very strong one, and it may also explain why the current system is more expensive that the one proposed by the authors: I expect that the current system is indeed based on a proper OD study of the city.
  2. Lines 152-157: How the authors have considered the effect of congestions? Especially when planning the model for the peak hours, and in cities that are already facing a significant congestion problem (and I assume the case study may be in a similar situation), the effect of traffic cannot be neglected. I don’t think it is reasonable to assume a speed of 30 km/h during the peak hour in a urban environment, as it would somewhat distort the results. This may be suitable in the case of reserved bus lanes, but I think it will be very difficult to implement them in an existing city, since this may be more similar to a BRT system.

Unfortunately, I think that these two limitations are very significant, and they may have a substantial impact on the results, especially when compared to the detailed analysis they performed for some phenomena. I believe that without a proper consideration of those hypotheses, the results of the authors may not represent in a reliable way the real operation of this system.

Additional minor aspects to be addressed:

The opportunity charging described at page 5 is not very clear to me. What impact would that have on the waiting time of the users at the stops? Would that be acceptable for them?

Line 377: Considering emission-free BEBs may overlook the impact of non-exhaust emissions, such as brakes, friction, tires and re-suspension effects. While it may be fine to exclude them from the analysis, I think it should be better to mention them.

Lines 542-543: Is the electricity mix of 2014 consistent with the future strategies of the country? Please further discuss this aspect, also considering the fact that a choice of BEBs may lead to their operation for many decades in the future.

Please check the quality of the English, since there are some typos in the manuscript (e.g. Line 116 “raw data” instead of “row data”. Line 121 “sensitivity analysis” instead of “sensitive analysis”. Line 398 “infraestructure”, etc.)

Author Response

Dear reviewer 3,

Thank you very much for your review. We have made the changes according to your comments. Please kindly check.

1) 

The transit network design problem has been analyzed from two different methodological approaches:

a) Discrete optimization techniques, based on OD matrices are able to resemble the connectivity and shape of the current street network and demand distribution. The resulting solutions in term of routes, stops and transit facilities are easily transferred to the arcs and edges that compose the network. Nevertheless, due to the high complexity nature of the discrete problem (NP-hard), the computational time of these solutions are extremely high in large networks. Moreover, the solutions and results obtained are site-dependent and cannot be transferred easily to other cities or implementation sites.

b) Alternatively to the aforementioned discrete approaches, continuous approximation models (CA) are robust techniques to design the desired skeleton of routes that the transit network should resemble. These methods are based of geometrical probability and only need a set of input parameters (economic and kinematic parameters) and total hourly demand, while some important assumptions should be made (uniform demand distribution, homogeneous transit vehicle technology, etc.).  The outstanding characteristics of these methods are based on the following issues:  transferability to multiple cities (we do not have to create the existing street network, we only need around 20 input parameters to describe the implementation site), fast calculation (solution calculated in less than 1 minute), and clear cause-effect relation among variables (it is not a black-box, the model is based on the formulation of transit performance variables as closed functions dependent to design variables). Despite of the severe assumptions,  continuous models  are robust and valid techniques, widely used to design the transit network. The contributions of this kind of models are published in top ranked journals:

Ansari, S., BaÅŸdere, M., Li, X., Ouyang, Y., & Smilowitz, K. (2018). Advancements in continuous approximation models for logistics and transportation systems: 1996–2016. In Transportation Research Part B: Methodological (Vol. 107, pp. 229–252). Elsevier Ltd. https://doi.org/10.1016/j.trb.2017.09.019

Badia, H., Estrada, M., Robusté, F., 2016. Bus network structure and mobility pattern: A monocentric analytical approach on a grid street pattern. Transportation Research Part B 93, 37-56.

Chen, H., Gu, W., Cassidy, M. J., & Daganzo, C. F. (2015). Optimal transit service atop ring-radial and grid street networks: A continuum approximation design method and comparisons. Transportation Research Part B: Methodological, 81, 755–774.

Daganzo, C.F., 2010. Structure of competitive transit networks. Transportation Research Part B 44 (4), 434-446.

Newell, G.F., 1979. Some issues relating to the optimal design of bus routes. Transportation Science 13 (1), 20-35.

Holroyd, E.M., 1967. The optimum bus service: a theoretical model for a large uniform urban area. In L. C. Edie, R. Herman, and R. Rothery (Eds.), Vehicular Traffic Science, In Proceedings of the 3rd International Symposium on the Theory of Traffic Flow. New York. Elsevier.

Nourbakhsh, S. M. and  Ouyang, Y., 2012. A structured flexible transit system for low demand areas. Transportation Research Part B 46(1), 204-216.

Wu, L., Gu, W., Fan, W., & Cassidy, M. J. (2020). Optimal design of transit networks fed by shared bikes. Transportation Research Part B: Methodological, 131, 63–83. https://doi.org/10.1016/j.trb.2019.11.003

Moreover, the major concern of this reviewer can be solved taking into account the achievements of the following paper, signed by one the authors of the paper under review. In the former paper, we validated and compared the cost and performance metrics estimated by continuous approximation models with the results obtained by simulation from the real OD matrix. We can conclude that the agreement of the two methodological approaches is within 85%. It means that, despite the strong assumptions, CA are valid techniques to identify the optimal design of transit networks with less than 15% of error in estimations. 

Estrada, M., Roca-Riu,M., Badia, H., Robusté, F. and  C Daganzo (2011). Design and implementation of efficient transit networks: Procedure, case study and validity testTransportation Research Part A: Policy and Practice 45 (9), 935-950

2) 

We assume that right of way measures are deployed along the layout of each route in the system. Therefore, bus motion is benefited from a segregated lane, that will result in a more stable commercial speed of buses. It has to be considered that the transfer-based grid network reduces by 25% the length of the transit service, in comparison to the old-fashioned existing design (Figure 3). Therefore, in comparison to the existing situation, we reduce the  length of the network along which priority measures have to be deployed. We agree that the more outstanding performance of the new bus design is achieved at the expense of deploying priority measures to ensure the conservative minimal threshold of cruising speed. This cruising speed (30 km/h) does not take into account dwell times, only traffic lights and other delays caused by traffic flow.

Round 2

Reviewer 1 Report

Dear Authors,

Thank You for the revision. #1 and #3 are solved from my previous review.

But issue #2 is just missed. I remind the issue 

#2

The results in the article are well presented. This is a strong element of the article but there is a lack of discussion. Thus please add a separate section with discussion.

I hope in the revised version I will be able to find this discussion section.

Best regards,

Reviewer

Author Response

(The authors gave the same response as above.)

Reviewer 3 Report

I am satisfied with the revision.

Author Response

Dear reviewer 3,

Thank you very much for your review. We have made the changes according to your comments in the revised version. Please kindly check.

Round 3

Reviewer 1 Report

Dear Authors,

Thank You for the deep revision. All my comments were included. So I recommend publishing this article in the revised form. 

I believe this article will find a deep audience in the scientific world.

With kind regards,

Reviewer